# Design and Synthesis of 1,3-Diarylpyrazoles and Investigation of Their Cytotoxicity and Antiparasitic Profile

**DOI:** 10.3390/ijms25094693

**Published:** 2024-04-25

**Authors:** Murat Bozdag, Freke Mertens, An Matheeussen, Natascha Van Pelt, Kenn Foubert, Nina Hermans, Guido R. Y. De Meyer, Koen Augustyns, Wim Martinet, Guy Caljon, Pieter Van der Veken

**Affiliations:** 1Laboratory of Medicinal Chemistry, Department of Pharmaceutical Sciences, Infla-Med Centre of Excellence, University of Antwerp, Universiteitsplein 1, 2610 Wilrijk-Antwerp, Belgiumkoen.augustyns@uantwerpen.be (K.A.); 2Laboratory of Physiopharmacology, Department of Pharmaceutical Sciences, Infla-Med Centre of Excellence, University of Antwerp, Universiteitsplein 1, 2610 Wilrijk-Antwerp, Belgium; freke.mertens@uantwerpen.be (F.M.); guido.demeyer@uantwerpen.be (G.R.Y.D.M.); wim.martinet@uantwerpen.be (W.M.); 3Laboratory of Microbiology, Parasitology and Hygiene, Department of Biomedical Sciences, Infla-Med Centre of Excellence, University of Antwerp, Universiteitsplein 1, 2610 Wilrijk-Antwerp, Belgium; an.matheeussen@uantwerpen.be (A.M.); natascha.vanpelt@uantwerpen.be (N.V.P.); guy.caljon@uantwerpen.be (G.C.); 4Natural Products and Food Research and Analysis–Pharmaceutical Technology, Department of Pharmaceutical Sciences, University of Antwerp, Universiteitsplein 1, 2610 Wilrijk-Antwerp, Belgium; kenn.foubert@uantwerpen.be (K.F.); nina.hermans@uantwerpen.be (N.H.)

**Keywords:** 1,3-diarylpyrazole, pyrazole, antiparasitic activity, cytotoxicity

## Abstract

Herein, we report a series of 1,3-diarylpyrazoles that are analogues of compound **26**/HIT 8. We previously identified this molecule as a ‘hit’ during a high-throughput screening campaign for autophagy inducers. A variety of synthetic strategies were utilized to modify the 1,3-diarylpyrazole core at its 1-, 3-, and 4-position. Compounds were assessed in vitro to identify their cytotoxicity properties. Of note, several compounds in the series displayed relevant cytotoxicity, which warrants scrutiny while interpreting biological activities that have been reported for structurally related molecules. In addition, antiparasitic activities were recorded against a range of human-infective protozoa, including *Trypanosoma cruzi*, *T. brucei rhodesiense*, and *Leishmania infantum*. The most interesting compounds displayed low micromolar whole-cell potencies against individual or several parasitic species, while lacking cytotoxicity against human cells.

## 1. Introduction

Among the *N*-heterocycles, pyrazoles are rarely present in natural products, but they are abundantly used in medicinal chemistry. They have been associated with a wide spectrum of biological activities [1,2,3]. Several clinically used drugs possess this pharmacophore, among which are the gastric secretion stimulant betazole, antibacterial drugs (e.g., sulfaphenazole), the anabolic steroid stanozolol, and non-steroidal anti-inflammatory drugs (NSAIDs; phenazone/antipyrine, benzydamine, tepoxalin, and deracoxib) [1,3,4,5,6,7,8]. In addition, the retinoic acid receptor agonist palovarotene and the hemoglobin modulator voxelotor contain pyrazole moieties [9,10]. They also share this feature with several protein kinase inhibitors (avapritinib, ruxolitinib, erdafitinib, crizotinib, and baricitinib) [11].

Moreover, the 1,3-diarylpyrazole substructure (which is investigated in this study) has been associated with a vast array of biological activities. Examples include lonazolac, which has found clinical use as a nonsteroidal anti-inflammatory drug (NSAID), and related compounds that are investigated as NSAIDs [12,13]. In addition, antimicrobial activities have been reported against Gram-positive bacteria, Gram-negative bacteria, fungi, and mycobacteria [14,15,16,17]. Other examples include inhibitors of histone H3K4 demethylase [18] and acetylcholinesterase [19]. Moreover, a recent study indicated the targeted anticancer properties of 1,3-diarylpyrazoles with substitution patterns at the 4-position [20].

Moreover, 1,3-diarylpyrazoles have been reported as antagonists of Toll-like receptor 9 (TLR9) [21]. TLRs are pattern recognition receptors located at the extracellular side of the membrane or on the luminal side of endosomes that initiate the innate immune response. They detect microbes and viruses via specific markers (lipoteichoic acids, lipoproteins, peptidoglycan, and single-stranded DNA) [22,23]. The group of Burger-Kentischer and Goldblum published, among others, compound **26**/HIT 8 (Figure 1) as a TLR9 antagonist. In this study, TLR9 affinity for compound **26**/HIT 8 was anticipated via in silico screening. An experimental EC50 of 3.16 μM for TLR9 antagonism was reported by these authors, relying on the readout of a cellular assay [21]. Of note, this study also reported compound **29** as an analogue with grossly comparable potency in the same cellular assay (Table 1, R = -CH_2_NHC_3_H_6_N(CH_3_)_2_).

Compound **26**/HIT 8 was also picked up by our groups as ‘HIT 8′ in a phenotypic high-throughput screening (HTS) for macro-autophagy inducers, referred to hereafter as ‘autophagy inducers’. More specifically, we screened a commercial drug-like compound library (Enamine; 10,240 compounds, unpublished results). The readout consisted of autophagosome counts in GFP-LC3 expressing L929 fibroblasts. Molecules were screened at 10 μM concentrations. Compound **26**/HIT 8 was found to be one of the most promising molecules in the screening. With the aim of further investigating the chemical space around this hit, a library of 57 analogues was synthesized that all share the 1,3-diarylpyrazole scaffold summarized in Table 1. Additionally, to further study the potential structural overlap between autophagy-inducing properties and TLR9 antagonism, we also prepared compound **29** (vide supra) [21]. We started with a cytotoxicity investigation of all diarylpyrazoles in a fibroblast cell line, namely, MRC-5. Surprisingly, this test did indicate that significant cytotoxicity is present for several of the diarylpyrazoles including the hit compound **26**/HIT 8 and its close analogue **29** in the same concentration range as used in the HTS test and the report by Burger-Kentischer and Goldblum [21]. This finding puts into question the validity of the published TLR9 data, because the phenotypic readout used in the manuscript indeed generates comparable readouts for TLR9 antagonism and cytotoxicity. Moreover, this finding also raises suspicion on the autophagy-inducing properties of **26**/HIT 8 that we observed during the HTS. Since autophagy is a cell-protective mechanism, it is not inconceivable that the observed increase was rather a cellular response against the toxicity of the molecule, as opposed to the effect of a direct interaction with the cell’s autophagic machinery. In the present study, we identify sub-classes of diarylpyrazole derivatives that have cytotoxicity risks. Finally, and probably most importantly, the observed cytotoxicity could raise potential questions on the validity of other published biological activities for 1,3-diarylpyrazoles: many of these have been published without cytotoxicity data. In that respect, the focus on cytotoxicity patterns for 1,3-diarylpyrazoles in this study can be of use to guide future drug discovery projects involving 1,3-diarylpyrazoles. Of note, for the compounds present in our 57-member library that we found to be non-toxic, we also report antiparasitic data against the disease-causing protozoa *Trypanosoma cruzi*, *T. brucei brucei*, *T. b. rhodesiense*, and *Leishmania infantum*. As mentioned earlier, 1,3-diarylpyrazoles have displayed promising activities against a wide variety of micro-organisms. The susceptibility of parasitic protozoa has—to the best of our knowledge—not been investigated yet. The selected micro-organisms are causative agents of, respectively, Chagas disease, sleeping sickness, and visceral leishmaniasis. These are diseases for which a high medical need persists today. For that reason, new structural classes endowed with activity against these parasites are dearly needed [24].

## 2. Results and Discussion

### 2.1. Chemistry

The generic structure of the 57 synthesized molecules is shown in Figure 2. These compounds share the overall 1,3-diarylpyrazole structure of HTS-hit **26**/HIT 8 and its close analogue **29**. Compared to the latter two, however, important variation was introduced at the R-position. A main rationale with this modification type was to investigate the influence of the basic side chains of **26**/HIT 8 and **29**. Because they are positively charged at physiological pH, they could have a relevant, negative effect on membrane permeability. To this end, we decided to replace these side chains with groups (a) either lacking basicity, (b) combining basicity with modified overall polarity, or (c) that have been reported in other series of biologically active analogues (Figure 1). In addition, we decided to investigate the importance of the N-aryl phenyl ring by replacing it with pyridine. As also illustrated by Figure 1, most reported biologically active diarylpyrazoles bear a phenyl ring at this position. Finally, the importance of the dioxane ring in **26**/HIT 8 and **29** was investigated by removing it or by replacing it with a methylenedioxy group or one or two methoxy substituents. A variety of different chemical strategies was utilized to synthesize these analogues. These are discussed separately in the following part.

First, the crucial building blocks **1**–**5** were prepared according to synthetic procedures from the literature [18] (Figure 1). First, 1-(2,3-dihydrobenzo[b][1,4]dioxin-6-yl)ethan-1-one was condensed with phenyl hydrazine in the presence of acetic acid to afford the corresponding hydrazone derivative. A Vilsmeier–Haack reaction was then used to convert the hydrazone to yield its 1,3-diarylpyrazole derivative **1**. In the same manner, analogues **2**–**5** were prepared, in which the 1,4-dioxane moiety was replaced by the corresponding 1,3-methylenedioxy derivative (**2**), the uncyclized 3,4-dimethoxyphenyl (**3**), or a 3-methoxyphenyl (**4**) derivative. In addition, the phenyl ring on the 1-position was manipulated by insertion of a nitrogen atom on its 2-position to yield the corresponding 2-pyridine derivative (**5**) (Figure 1).

From there, the first synthetic strategy to build up the library of 1,3-diarylpyrazoles involved reductive amination of the aldehydes (**1**–**5**) with a series of amine derivatives. (Figure 2, Entry A) (Figure 2). This series covered key molecules **26**/HIT 8 and the analogous **29** along with analogues with a diversity-driven set of aliphatic and aromatic amine substituents (compounds **13**–**29**, **55**–**61**, **62**–**66**, **67**, and **69**). The structures of these compounds are provided in Table 1. Furthermore, a second synthetic strategy was focused on obtaining amide analogues of **26**/HIT 8. (Figure 2, Entry B) Thus, the aldehyde derivative (**1**) was oxidized by potassium permanganate to afford the corresponding carboxylic acid derivative (**6**), which was then reacted with a series of amines in the presence of a coupling reagent to afford a series of amide-based final products with aromatic and aliphatic tails with terminal aromatic moieties (**32**–**37**), as well as aliphatic tails (**39**–**44**) (Figure 2).

The third derivatization strategy consisted of the reduction in the aldehyde moiety on the 4-position of the 1,3-diarylpyrazole molecules **1** and **4** to obtain the corresponding alcohol derivatives **7** and **8**. The latter were then reacted with 4-fluorobenzyl bromide in the presence of sodium hydride to obtain the corresponding benzyl ether derivatives (**45** and **68**). Conversely, treatment of the alcohol derivative **7** with propargyl bromide in the presence of a base resulted in propargyl ether derivative **9**, which was treated with phenyl or benzyl azides in ‘click chemistry’ conditions to afford ether derivatives with phenyl or benzyl triazole tails (**46**–**48**) (Figure 3).

Our fourth strategy was focused on the deletion of the heteroatom at the 4-position of compound **26**/HIT 8. (Figure 4) Thus, metal-free C–C bond formation was affected between the aldehyde 1 and a boronic acid to afford derivatives **49**–**51**, according to strategy from the literature based on tosylhydrazone cross-coupling [25,26].

To further enrich the chemical features of the 1,3-diarylpyrazole series, three additional molecules, **52**, **53**, and **54**, (Figure 5 and Figure 6) were synthesized. These truncated analogues served to assess the importance of the substituent at the 4-position for cytotoxicity and biological activity. At the same time, these fluorinated molecules were expected to possess potentially optimized biopharmaceutical properties compared to the parent compound **26**/HIT 8 (e.g., increased cellular permeability and/or metabolic stability).

For their preparation, the 4-formyl derivative **1** and its reduced 4-hydroxymethyl analogue **7** were used in nucleophilic fluorination reactions with DAST to afford 4-difluoromethyl and 4-fluoromethyl derivatives **53** and **52**, respectively, according to the previously reported literature for fluorination of 4-position of 1,3-diarylpyrazoles [27]. Finally, the 4-aminomethyl derivative **12** was also synthesized starting from the 4-alcohol derivative **2** in three steps according to the previously reported literature [28]. The synthesis was carried out by treatment of derivative **7** with tosyl chloride to obtain the tosylate **10**. The latter was transformed into its azide derivative **11** by sodium azide and directly reduced with lithium aluminum hydride to afford the 4-aminomethyl intermediate **12**. This compound was used without any further purification and treated with 4-fluorophenyl isocyanate to afford the corresponding ureido derivative **54** (Figure 6).

### 2.2. Cytotoxicity and Antiparasitic Activity

The series of final products (**13**–**69**) was investigated in vitro on the human fetal lung fibroblast cell line MRC-5 (Table 1) to assess cytotoxicity toward normal (i.e., non-malignant) human cells. For compounds lacking MRC-5 cytotoxicity (EC_50_ > 64 µM, the highest concentration evaluated) and compounds with limited cytotoxicity (20 µM < EC_50_ < 64 µM), activities against four parasitic protozoans (*T. cruzi*, *L. infantum*, *T. b. brucei*, and *T. b. rhodesiense*) were also investigated. As explained earlier, these are important pathogens for humans or cattle (*T. b. brucei*) and a persistent need for new therapeutics exists for these parasites. As a part of the antiparasitic tests, cytotoxicity against primary mouse macrophages (PMMs) was also evaluated. These murine cells serve as host cells for parasites in the *L. infantum* assay. Therefore, the murine macrophage assay not only offers additional cytotoxicological insight but is also essential to interpreting the readout of the *L. infantum* assay.

In total, 21 of the 57 compounds tested displayed strong cytotoxicity in MRC-5 cells (0.56 µM < EC50 < 20 μM). In contrast, 18 molecules caused no measurable cytotoxic response (EC50 > 64 µM). Eighteen compounds were found to have measurable but limited cytotoxicity in the assay (20 µM < EC50 < 64 µM). When trying to map the structure toxicity range for these compounds, some observations can be made. Most importantly, the nature of the ‘R’ side chain seems to play a decisive role in the presence or absence of toxicity, implying that the 1,3-diarylpyrazole basic structure of the compounds is not an inherently problematic substructure. This in itself is a relevant finding, given the high number of biologically active 1,3-diarylpyrazoles that have been reported. Practically all toxic compounds in the series contain a basic, *N*-substituted aminomethyl R-group. Remarkably, when combined with a phenylethyl *N*-substituent, the basic aminomethyl R-group seems to lead to more toxic representatives than—for example—with benzyl-derived *N*-substituents. This is reflected by comparing the most toxic (phenylethyl-derived) molecule in the series (**17**) with its benzyl analogues **15** and **16**: the latter two are also toxic, but an order of magnitude less than their homologous counterpart. This remains the case when studying compounds that have a methylenedioxy- or methoxy-based X-part (compounds **55**, **56**, and **67**). Replacing the phenyl ring in these compounds with a pyridine generally lowers toxicity by around an order of magnitude but does not lead to molecules with a ‘safe’ profile. This is shown by compounds **19**–**21**, **57**–**59**, and **62**–**64**, with the latter two subgroups again being representatives of the methylenedioxy- and methoxy-based X-series. The presence of an *N*-alkyl substituent on the aminomethyl moiety also holds a cytotoxicity risk, which is bigger with lipophilic alkyl substituents. The *N*-adamantyl-substituted **25** illustrates this, as does the consistently lower cytotoxicity of molecules with a polar N-alkyl group (e.g., **23**–**24** and **26/HIT 8**–**29**). Finally, only one type of *N*-substituent on the aminomethyl moiety seems to lead to compounds with a reduced toxicity risk: *N*-aryl groups (**13**, **14**, **18**, and **22**). An appealing explanation for this finding could be that these arylamines have significantly lower basicity and are not protonated at physiological pH. Indicative of the same critical contribution of a basic aminomethyl R-group to cytotoxicity are the molecules in which this moiety has been replaced by a non-basic amide function (**33**–**37** and **39**–**44**), an ether group (**45–48** and **68**), or aliphatic or aromatic R-groups (**49**–**53**). In addition, the non-basic ureido derivative **54** also fits in this series. All of these modifications clearly improve the cytotoxicity profile, although some representatives still have moderately potent EC_50_ values (20–64 µM) in MRC-5 cells. A notable, toxic outlier is again the phenylethyl-containing amide product **34**.

Overall, these cytotoxicity results put into question the effects of these molecules observed in our own autophagy screening and in the reported TLR9 assays because phenotypic EC_50_ values, for both molecules, are well within the cytotoxic range. Further biological research on the mechanism of toxicity would be required. A broad screening against protein kinases could, for example, already shed light on whether affinity toward these master regulators in cellular physiology is involved in the observed toxicity. Moreover, these findings raise scrutiny when interpreting the biological activities in many other publications that report, e.g., antibiotic properties or cholinesterase inhibition by structurally related 1,3-diarylpyrazoles, but with no cytotoxicity data (shown in Figure 1).

Finally, we submitted compounds with MRC-5 EC_50_s > 20 µM to the antiparasitic screening mentioned earlier (results also shown in Table 1). Of note, the *N*-aryl-substituted **13** and **14** were found to be active against *T. cruzi* (EC_50_ of 2.23 μM) while **33** and **44** were active against *L. infantum* (EC_50_ 8.30 and 9.51 μM, respectively) and **22**, **35**, **38**, and **48** showed potency against *T. b. rhodesiense* (EC_50_ spanning between 0.74 and 1.78 μM). Several compounds (**14**, **21**, **26**/HIT 8, and **41**) were identified with modest but broad antiparasitic activity (EC_50_ < 10 µM) against all tested pathogens. While several of these results would justify the synthesis of further analogues, the risk for identifying cytotoxic molecules is currently restraining us from doing so.

## 3. Materials and Methods

### 3.1. Chemistry

Reagents were obtained from commercial sources and were used without further purification. Characterization of all compounds was carried out with ^1^H and ^13^C NMR and mass spectrometry. ^1^H and ^13^C NMR spectra were recorded on a 400 MHz Bruker Avance III Nanobay spectrometer with Ultrashield working at 400 and 100 MHz, respectively, and analyzed by use of MestReNova analytical chemistry software version 14.1.1-24571. Chemical shifts are in ppm, and coupling constants are in hertz (Hz). Splitting patterns are designated by s (singlet), d (doublet), t (triplet), q (quartet), m (multiplet), brs (broad singlet), and dd (double of doublets). The UPLC (ultraperformance liquid chromatography) used to quantify the purity of the products was an ACQUITY UPLC H-Class system with a TUV detector Waters coupled to an MS detector Waters QDa. An Acquity UPLC BEH C18 1.7 μm (2.1 mm × 50 mm) column was used and as eluent with a mixture of 0.1% FA in H_2_O, 0.1% FA in ACN, H_2_O, and ACN. The wavelengths for UV detection were 254 and 214 nm. When necessary, flash column chromatography was performed on a Biotage ISOLERA One flash system equipped with an internal variable dual-wavelength diode array detector (200–400 nm). For normal phase purifications, Biotage Sfär cartridges (5–100 g, flow rate of 10–100 mL/min) were used, and reverse phase purifications were performed, making use of Büchi C18 cartridges (4–30 g, flow rate of 10–50 mL/min). Dry sample loading was carried out by self-packing sample cartridges using Celite 545. The gradients used varied for each purification.

The following sections comprise the synthetic procedures and analytical data for all compounds reported in this manuscript. Several synthetic procedures that were used in the preparation of intermediates and final products are summarized here as ‘General Procedures’. Target compounds were obtained with a purity of >95%. Most of the compounds described in this paper have been synthesized according to general procedures A–H.

*General procedure A.* General procedure A describes the synthesis of aldehyde derivatives 1–5 (Figure 1). A solution of acetophenone (a1–a4) (1 equiv.) in EtOH (70 mL) was treated with acetic acid (1.2 equiv.) and hydrazine (b1–b2) (1.2 equiv.). The reaction was left to stir until the consumption of the starting materials (TLC/UPLC monitoring). The obtained precipitates were filtered off and washed with cold EtOH (3 × 20 mL) to afford hydrazones (c2–c4), or the solvents were removed under reduced pressure to obtain the hydrazones (c1, c5) to use without further purification. In a round-bottom flask, phosphorus oxychloride (3.0 equiv.) was cooled to 0 °C; then, previously cooled (0–5 °C) DMF (3.0 equiv.) was added dropwise and left to stir for 15–20 min to form so-called Vilsmeier reagent (chloroiminium ion). A solution of hydrazone (c1–c5) (0.01 mol, 1 equiv.) in DMF (15–20 mL) was added dropwise to the reaction mixture using an addition funnel and heated at 80 °C for 3 h. After the consumption of the starting materials (TLC monitoring), the mixture was cooled to 0 °C, quenched with slush, and neutralized with NaOH. The precipitate was filtered off, washed with water (3 × 30 mL) to afford aldehydes (1–5), and used without any further purification.

*General Procedure B*. General Procedure B outlines the reductive amination reactions in Figure 2 to afford compounds 15–17, 19–21, 23–29, 55–59, 61–67 and 69. A solution of aldehyde (1–5) (0.25 g, 1.0 equiv.) in 5 mL dry EtOH was treated with the appropriate primary or secondary aliphatic amine (1.1 equiv.) at the room temperature. The reaction was stirred for four hours, then cooled-down to 0 °C and sodium borohydride (4.0 equiv.) was added portion-wise, and allowed to reach rt. The consumption of the starting materials was monitored by UPLC and then quenched with slush and basified by saturated solution of sodium bicarbonate. The solvents were removed under reduced pressure and the obtained residue extracted with ethyl acetate (3 × 10 mL). The combined organic layers were washed with water (3 × 30 mL) and dried over Na_2_SO_4_, and the solvents were removed under reduced pressure to give a residue that was then purified by silica gel/reverse-phase flash column chromatography. The combined pure fractions were evaporated under vacuum to give the products, which were converted into their corresponding hydrochloride (15–17, 19–21, 24–29, 55–59, 61–64, 66–67, and 69) or oxalate salts (23, 65). Subsequentely, they were dried in a vacuum oven (40 °C) to obtain the titled products as amorphous solids.

*General Procedure C*. General procedure C outlines the reductive amination reactions in Figure 2 to afford amine derivatives 13–14, 18, 22, and 60. A mixture of aldehyde 1–2 (0.25 g, 1.0 equiv.) and appropriate aniline derivative (1.0 equiv.) in 5 mL dry THF was treated with acetic acid (0.1 equiv.) and then cooled down to 0 °C, followed by addition of sodium cyanoborohydride (2.5 equiv.). The reaction was allowed to stir at rt until the consumption of the starting materials (UPLC monitoring) and then quenched with slush, pH was basified by adding 5 mL of aqueous sodium bicarbonate saturated solution, extracted with ethyl acetate (3 × 10 mL). The combined organic layers were washed with water (2 × 15 mL), brine (1 × 15 mL) and dried over Na_2_SO_4_, and solvents removed under reduced pressure to give a residue which was purified by silica gel/reverse-phase flash column chromatography. The combined pure fractions were evaporated under vacuum and subsequently dried under high vacuum for 5 h to to give titled compounds 13–14, 18, 22, 60 as amorphous solids.

*General Procedure D*. General procedure D outlines the synthesis of amide derivatives 30–44 reported in Figure 2. A solution of carboxylic acid derivative 6 (1.0 equv.) in dry DMF at 0 °C was treated with *N*-hydroxy succinimide (1.2 equiv.) and EDCl (1.2 equiv.) (for the aliphatic amine derivatives) or HATU (1.2 equiv.) (for the aniline derivatives 30–31 and 38) and stirred for 30 min, followed by the addition of an appropriate amine (1.0 equiv.), DIPEA (2.5 equiv.) and allowed to warm to rt. The reaction was continued until the consumption of the starting materials (UPLC monitoring), then quenched with water, and extracted with ethyl acetate (3 × 10 mL); the combined organic layers were washed with water (2 × 15 mL) and brine (1 × 15 mL) and dried over Na_2_SO_4_, and solvents were removed under reduced pressure to give a residue that was purified by silica gel/reverse-phase flash column chromatography. The combined pure fractions were evaporated under vacuum and subsequently dried under high vacuum for 5 h to afford the titled compounds 30–44 as amorphous solids.

*General Procedure E.* General procedure E outlines the preparation of ether derivatives 9, 45, and 68 reported in Figure 3. A solution of the aldehyde derivative 1, 4 (1.0 equiv.) in MeOH was treated with sodium borohydride (4.0 equiv.) and warmed to rt. The reaction was quenched with slush after the consumption of the starting materials (UPLC monitoring). Excess solvents were removed under reduced pressure and extracted with ethyl acetate (3 × 10 mL); the combined organic layers were washed with water (2 × 15 mL) and brine (1 × 15 mL) and dried over Na_2_SO_4_, and solvents were removed under reduced pressure to give the alcohol derivatives 7–8. A solution of alcohol derivative 7–8 (1.0 equiv.) in 5 mL of dry THF was cooled down to −10 °C, treated with sodium hydride (5.0 equiv.), and left to stir for 30 min, followed by the addition of alkyl bromide (1.2 equiv.), and then warmed to rt. The reaction was continued until the consumption of the starting materials (UPLC monitoring), quenched with slush, and extracted with ethyl acetate (3 × 10 mL), and the combined organic layers were washed with water (2 × 15 mL) and brine (1 × 15 mL) and dried over Na_2_SO_4_, and solvents were removed under reduced pressure to give a residue that was used as such for compound 9 or purified by flash chromatography for 45 and 68. The combined pure fractions were evaporated under vacuum and subsequently dried under high vacuum for 5 h to afford the titled compounds 45 and 68 as amorphous solids. 

*General procedure F.* General procedure F outlines the synthesis of triazole derivative 46–48 reported in Figure 3. A suspension of the alkyne 9 (1.0 equiv.) and the azide derivative d1–d3 (1.5 equiv.) in 5.0 mL of *tert*-butanol/water (1:1) was treated with sodium ascorbate (0.5 equiv.) and copper (II) sulfate (0.1 equiv.) and left to stir on. The reaction was quenched with water and extracted with EtOAc (3 × 15 mL); the combined organic layers were dried over Na_2_SO_4_ and the solvents were evaporated under reduced pressure to obtain a crude that was purified by flash column chromatography. The combined pure fractions were evaporated under vacuum and subsequently dried under high vacuum for 5 h to afford the compounds 46–48 as amorphous solids.

*General procedure G.* General procedure G outlines the metal-free synthesis of cross-coupling products 49–51 reported in Figure 4. A solution of aldehyde 1 (1.0 equiv.) and tosylhydrazide (1.0 equiv.) in 5 mL of 1,4-dioxane was heated to 80 °C for 2 h in a sealed tube; then, the mixture was cooled down to rt, and potassium carbonate (1.5 equiv.) and phenyl boronic acid e1–e3 (1.5 equiv.) were added to the mixture and heated to 110 °C until the consumption of the starting materials (UPLC monitoring). The reaction was quenched with water; solvents were removed under reduced pressure to obtain a crude that was purified by flash chromatography. The combined pure fractions were evaporated under vacuum and subsequently dried under high vacuum for 5 h to afford products 49–51 as amorphous solids.

*General procedure H.* General procedure H outlines the synthesis of products 52–53 by nucleophilic fluorination reactions reported in Figure 5. A solution of the aldehyde derivative 1 and the alcohol derivative 7 (1.0 equiv.) in 10 mL DCM was treated with DAST (3.5–6.5 equiv.) dropwise at −10 °C and left to warm to rt. The reaction was stirred until the consumption of the starting materials (UPLC monitoring), quenched with a saturated solution of sodium bicarbonate, and extracted with DCM (3 × 10 mL). The combined organic layers were washed with water (3 × 10 mL) and dried over Na_2_SO_4_, and solvents were removed under reduced pressure to obtain a residue that was purified by flash chromatography on silica gel with EtOAc in *n*-heptane (10–80%). The combined pure fractions were evaporated under vacuum and subsequently dried under high vacuum for 5 h to afford the title compounds **53** and **52** as amorphous solids, respectively.


***3-(2,3-dihydrobenzo[b][1,4]dioxin-6-yl)-1-phenyl-1H-pyrazole-4-carbaldehyde* (1).**


Brown Solid, 60% yield. ^1^H NMR (400 MHz, CDCl_3_) δ 10.04 (s, 1H), 8.51 (s, 1H), 7.78 (d, *J* = 8.4 Hz, 2H), 7.50 (d, *J* = 8.3 Hz, 2H), 7.44–7.35 (m, 2H), 7.32 (dd, *J* = 8.3, 2.1 Hz, 1H), 6.99 (d, *J* = 8.3 Hz, 1H), 4.32 (s, 4H); ^13^C NMR (101 MHz, CDCl_3_) δ 184.7, 154.3, 144.8, 143.8, 139.1, 130.6, 129.7, 127.9, 124.8, 122.4, 122.3, 119.7, 118.0, 117.7, 64.6, 64.4. UPLC/MS (ESI): *m*/*z* = 306.9 [M+H]^+^.


***3-(benzo[d][1,3]dioxol-5-yl)-1-phenyl-1H-pyrazole-4-carbaldehyde* (2) [29].**


Yellow solid, 43% yield. ^1^H NMR (400 MHz, DMSO) δ 9.95 (s, 1H), 9.31 (s, 1H), 7.98 (d, *J* = 7.6 Hz, 2H), 7.62–7.53 (m, 2H), 7.53–7.48 (m, 2H), 7.42 (t, *J* = 7.6 Hz, 1H), 7.05 (d, *J* = 8.7 Hz, 1H), 6.11 (s, 2H). ^13^C NMR (101 MHz, DMSO) δ 184.3, 153.2, 149.1, 148.4, 139.0, 136.2, 130.7, 128.6, 126.0, 123.9, 122.9, 120.1, 109.7, 109.3, 102.3. UPLC/MS (ESI): *m*/*z* = 293.1 [M+H]^+^.


***3-(3,4-dimethoxyphenyl)-1-phenyl-1H-pyrazole-4-carbaldehyde* (3).**


Beige Solid, 55% yield, ^1^H NMR (400 MHz, DMSO) δ 9.98 (s, 1H), 9.30 (s, 1H), 8.00 (d, *J* = 7.4 Hz, 2H), 7.61–7.51 (m, 4H), 7.42 (t, *J* = 7.4 Hz, 1H), 7.08 (d, *J* = 8.4 Hz, 1H), 3.84 (s, 3H), 3.82 (s, 3H); ^13^C NMR (101 MHz, DMSO) δ 185.7, 153.6, 150.7, 149.5, 139.6, 136.0, 130.6, 128.6, 124.7, 123.0, 122.4, 120.2, 113.0, 112.5, 56.5.


***3-(2,3-dihydrobenzo[b][1,4]dioxin-6-yl)-1-phenyl-1H-pyrazole-4-carboxylic acid* (6).**


A suspension of compound 1 (2.0 g, 1.0 equiv.) in 20 mL water/pyridine (1:1) mixture was treated with potassium permanganate (1.0 equiv.) portion-wise. The reaction mixture was stirred until the disappearance of the violet color; then, the manganese dioxide precipitate was filtered off and washed with 1M aqueous solution of sodium hydroxide. This was followed by acidification of the solution by 1M aqueous HCl solution to obtain a precipitate that was filtered off, washed with water (3 × 30 mL), and dried under vacuum to afford the titled compound **6**.

Beige solid. 77% yield, ^1^H NMR (400 MHz, DMSO) δ 12.57 (brs, 1H, COO*H*), 9.03 (s, 1H), 7.96 (d, *J* = 7.7 Hz, 2H), 7.53 (t, *J* = 7.7 Hz, 2H), 7.42 (d, *J* = 2.0 Hz, 1H), 7.40–7.33 (m, 2H), 6.91 (d, *J* = 8.4 Hz, 1H), 4.28 (s, 4H); ^13^C NMR (101 MHz, DMSO) δ 164.8, 153.1, 144.8, 143.6, 139.7, 134.6, 130.6, 128.1, 126.1, 123.1, 119.9, 118.9, 117.4, 114.5, 65.2, 65.0. UPLC/MS (ESI): *m*/*z* = 323.1 [M+H]^+^.


***(3-(2,3-dihydrobenzo[b][1,4]dioxin-6-yl)-1-phenyl-1H-pyrazol-4-yl)methanol* (7).**


Brown solid, 85% yield; ^1^H NMR (400 MHz, CDCl_3_) δ 7.98 (s, 1H), 7.73 (d, *J* = 7.6 Hz, 2H), 7.45 (t, *J* = 7.6 Hz, 2H), 7.41 (d, *J* = 2.1 Hz, 1H), 7.36 (dd, *J* = 8.3, 2.1 Hz, 1H), 7.29 (m, 1H), 6.95 (d, *J* = 8.3 Hz, 1H), 4.76 (s, 2H), 4.30 (s, 4H); ^13^C NMR (101 MHz, CDCl_3_) δ 151.2, 143.9, 143.8, 140.0, 129.6, 127.8, 126.5, 121.2, 120.6, 119.0, 117.6, 116.9, 64.6, 64.5, 56.2. UPLC/MS (ESI): *m*/*z* = 309.1 [M+H]^+^.


***(3-(4-methoxyphenyl)-1-phenyl-1H-pyrazol-4-yl)methanol* (8).**


Brown solid, 75% yield, ^1^H NMR (400 MHz, DMSO) δ 8.48 (s, 1H), 7.88 (d, *J* = 7.4 Hz, 2H), 7.84 (d, *J* = 8.8 Hz, 2H), 7.50 (t, *J* = 7.4 Hz, 2H), 7.29 (t, *J* = 7.4 Hz, 1H), 7.04 (d, *J* = 8.8 Hz, 2H), 5.20 (brs, 1H), 4.53 (s, 2H), 3.81 (s, 3H); ^13^C NMR (101 MHz, DMSO) δ 159.6, 150.8, 140.5, 130.5, 129.6, 129.5, 127.3, 126.5, 122.0, 118.9, 114.9, 56.1, 55.1.


***(3-(2,3-dihydrobenzo[b][1,4]dioxin-6-yl)-1-phenyl-1H-pyrazol-4-yl)methyl 4-methylbenzenesulfonate* (10).**


White solid, 55% yield, ^1^H NMR (400 MHz, DMSO) δ 8.74 (s, 1H), 7.93 (d, *J* = 7.7 Hz, 2H), 7.60–7.51 (m, 2H), 7.47 (d, *J* = 8.1 Hz, 2H), 7.39 (t, *J* = 7.7 Hz, 1H), 7.14–7.09 (m, 3H), 7.07 (dd, *J* = 8.2, 2.0 Hz, 1H), 7.00 (d, *J* = 8.2 Hz, 1H), 4.43 (s, 2H), 4.30 (s, 4H), 2.28 (s, 3H); ^13^C NMR (101 MHz, DMSO) δ 154.4, 146.8, 144.9, 144.5, 139.9, 138.5, 132.7, 130.5, 129.0, 128.0, 126.4, 126.1, 123.0, 119.8, 118.6, 118.4, 108.3, 65.2, 65.0, 51.1, 21.7.


***N-((3-(2,3-dihydrobenzo[b][1,4]dioxin-6-yl)-1-phenyl-1H-pyrazol-4-yl)methyl)aniline* (13).**


Pale yellow solid, 8% yield. ^1^H NMR (400 MHz, DMSO) δ 8.54 (s, 1H), 7.86 (d, *J* = 7.7 Hz, 2H), 7.56–7.46 (m, 2H), 7.36–7.24 (m, 3H), 7.10 (d, *J* = 7.6 Hz, 2H), 6.93 (d, *J* = 8.9 Hz, 1H), 6.67 (d, *J* = 7.6 Hz, 2H), 6.58 (t, *J* = 7.6 Hz, 1H), 5.91 (t, *J* = 4.9 Hz, 1H, N*H*), 4.26 (s, 4H), 4.18 (d, *J* = 4.9 Hz, 2H); ^13^C NMR (101 MHz, DMSO) δ 151.1, 149.6, 144.4, 144.3, 140.4, 130.5, 129.9, 129.8, 127.1, 127.0, 121.4, 119.4, 118.9, 118.2, 117.1, 116.9, 113.3, 65.1, 65.0, 39.1. UPLC/MS (ESI): *m/z* = 384.3 [M+H]^+^.


***N-((3-(2,3-dihydrobenzo[b][1,4]dioxin-6-yl)-1-phenyl-1H-pyrazol-4-yl)methyl)-4-fluoroaniline* (14).**


Pale yellow solid 7% yield. ^1^H NMR (400 MHz, DMSO) δ 8.55 (s, 1H), 7.86 (d, *J* = 8.2 Hz, 2H), 7.50 (t, *J* = 8.2 Hz, 2H), 7.36–7.24 (m, 3H), 7.01–6.89 (m, 3H), 6.72–6.60 (m, 2H), 5.87 (t, *J* = 5.0 Hz, 1H, N*H*), 4.26 (m, 4H), 4.15 (d, *J* = 5.0 Hz, 2H); ^13^C NMR (101 MHz, DMSO) δ 155.5 (d, ^1^*J*_C–F_ = 230 Hz), 151.1, 146.3 (d, ^4^*J*_C–F_ = 2 Hz), 144.3 (d, ^3^*J*_C–F_ = 8 Hz), 140.4, 130.5, 129.9, 127.1, 127.0, 121.4, 119.4, 118.9, 118.2, 117.0, 116.2 (d, ^2^*J*_C–F_ = 22 Hz), 114.1, 114.1, 65.1, 65.0, 39.7. UPLC/MS (ESI): *m/z* = 402.3 [M+H]^+^.


***N-benzyl-1-(3-(2,3-dihydrobenzo[b][1,4]dioxin-6-yl)-1-phenyl-1H-pyrazol-4-yl)methanamine hydrochloride* (15).**


White solid, 35% yield, ^1^H NMR (400 MHz, DMSO) δ 9.44 (s, 2H, N*H*_2_), 8.75 (s, 1H), 7.82 (d, *J* = 8.0 Hz, 2H), 7.61–7.48 (m, 4H), 7.48–7.40 (m, 3H), 7.37 (t, *J* = 8.0 Hz, 1H), 7.15 (d, *J* = 2.1 Hz, 1H), 7.07 (dd, *J* = 8.4, 2.1 Hz, 1H), 6.93 (d, *J* = 8.4 Hz, 1H), 4.30 (s, 4H), 4.26–4.19 (m, 4H); ^13^C NMR (101 MHz, DMSO) δ 152.0, 144.6, 144.4, 140.2, 132.7, 131.3, 131.2, 130.7, 129.8, 129.5, 127.7, 126.1, 122.1, 119.3, 118.2, 117.7, 112.8, 65.2, 65.0, 50.5, 41.3. UPLC/MS (ESI): *m/z* = 398.3 [M+H]^+^.


***1-(3-(2,3-dihydrobenzo[b][1,4]dioxin-6-yl)-1-phenyl-1H-pyrazol-4-yl)-N-(4-fluorobenzyl)methanamine hydrochloride* (16).**


Pale brown solid, 55% yield. ^1^H NMR (400 MHz, DMSO) δ 9.70 (s, 2H, N*H*_2_), 8.85 (s, 1H), 7.83–7.77 (m, 2H), 7.61 (m, 2H), 7.55 (m, 2H), 7.40–7.33 (m, 1H), 7.26 (t, *J* = 8.9 Hz, 2H), 7.11 (d, *J* = 2.0 Hz, 1H), 7.06 (dd, *J* = 8.3, 2.1 Hz, 1H), 6.92 (d, *J* = 8.3 Hz, 1H), 4.29 (s, 4H), 4.22 (s, 2H), 4.17 (s, 2H); ^13^C NMR (101 MHz, DMSO) δ 163.3 (d, ^1^*J*_C-F_ = 245 Hz), 151.9, 144.7, 144.4, 140.1, 133.5 (d, ^3^*J*_C-F_ = 8 Hz), 131.2, 130.7, 129.0, 127.7, 126.0, 122.1, 119.3, 118.2, 117.6, 116.4 (d, ^2^*J_C-F_* = 21 Hz), 112.8, 65.2, 65.0, 49.8, 41.3. UPLC/MS (ESI): *m/z* = 416.3 [M+H]^+^.


***N-((3-(2,3-dihydrobenzo[b][1,4]dioxin-6-yl)-1-phenyl-1H-pyrazol-4-yl)methyl)-2-phenylethan-1-amine hydrochloride* (17).**


White solid, 85% yield. ^1^H NMR (400 MHz, DMSO) δ 9.33 (s, 2H, N*H*_2_), 8.83 (s, 1H), 7.81 (d, *J* = 8.0 Hz, 2H), 7.56 (t, *J* = 8.0 Hz, 2H), 7.40–7.29 (m, 3H), 7.29–7.21 (m, 3H), 7.21–7.12 (m, 2H), 6.98 (d, *J* = 8.2 Hz, 1H), 4.30 (s, 4H), 4.24 (m, 2H), 3.22 (m, 2H), 2.99 (m, 2H); ^13^C NMR (101 MHz, DMSO) δ 151.8, 144.7, 144.4, 140.1, 138.0, 131.1, 130.7, 129.6, 129.5, 127.7, 126.1, 122.1, 119.3, 118.3, 117.6, 112.9, 65.2, 65.1, 48.5, 41.9, 32.4. UPLC/MS (ESI): *m/z* = 412.3 [M+H]^+^.


***5-(((3-(2,3-dihydrobenzo[b][1,4]dioxin-6-yl)-1-phenyl-1H-pyrazol-4-yl)methyl)amino)-N-methylpicolinamide* (18).**


Pale yellow solid, 8% yield. ^1^H NMR (400 MHz, DMSO) δ 8.58 (s, 1H), 8.31 (q, *J* = 5.0 Hz, 1H, N*H*), 8.04 (d, *J* = 2.7 Hz, 1H), 7.86 (d, *J* = 8.0 Hz, 2H), 7.78 (d, *J* = 8.5 Hz, 1H), 7.51 (t, *J* = 8.0 Hz, 2H), 7.39–7.19 (m, 3H), 7.09 (dd, *J* = 8.5, 2.7 Hz, 1H), 6.93 (d, *J* = 8.3 Hz, 1H), 6.84 (t, *J* = 4.9 Hz, 1H, N*H*), 4.28 (m, 6H), 2.76 (d, *J* = 5.0 Hz, 3H); ^13^C NMR (101 MHz, DMSO) δ 165.8, 151.1, 147.4, 144.4, 144.3, 140.6, 139.1, 134.6, 130.5, 130.0, 127.1, 126.9, 123.6, 121.4, 119.0, 118.4, 118.2, 118.1, 116.9, 65.1, 65.0, 38.5, 26.7. UPLC/MS (ESI): *m/z* = 442.3 [M+H]^+^.


***N-((3-(2,3-dihydrobenzo[b][1,4]dioxin-6-yl)-1-phenyl-1H-pyrazol-4-yl)methyl)-2-(pyridin-4-yl)ethan-1-amine hydrochloride* (19).**


Brown solid, 68% yield. ^1^H NMR (400 MHz, DMSO) δ 9.95 (s, 2H, N*H_2_*), 9.02 (s, 1H), 8.86 (d, *J* = 5.9 Hz, 2H), 7.99 (d, *J* = 5.9 Hz, 2H), 7.78 (d, *J* = 7.9 Hz, 2H), 7.55 (t, *J* = 7.9 Hz, 2H), 7.36 (t, *J* = 7.9 Hz, 1H), 7.24–7.11 (m, 2H), 6.97 (d, *J* = 8.1 Hz, 1H), 4.30 (s, 4H), 4.21 (m, 2H), 3.48–3.31 (m, 4H); ^13^C NMR (101 MHz, DMSO) δ 151.8, 144.7, 144.4, 143.1, 140.1, 131.6, 131.2, 130.7, 128.1, 127.7, 126.1, 122.2, 119.2, 118.3, 117.7, 112.8, 65.1, 65.0, 46.4, 42.0, 32.2. UPLC/MS (ESI): *m/z* = 413.3 [M+H]^+^.


***N-((3-(2,3-dihydrobenzo[b][1,4]dioxin-6-yl)-1-phenyl-1H-pyrazol-4-yl)methyl)-2-(pyridin-3-yl)ethan-1-amine hydrochloride* (20).**


White solid, 35% yield. ^1^H NMR (400 MHz, DMSO) δ 9.80 (brs, 2H, N*H*_2_), 9.00 (m, 1H), 8.89 (s, 1H), 8.79 (d, *J* = 5.6 Hz, 1H), 8.45 (m, 1H), 7.95 (m, 1H), 7.79 (d, *J* = 8.0 Hz, 2H), 7.56 (t, *J* = 8.0 Hz, 2H), 7.37 (t, *J* = 8.0 Hz, 1H), 7.21–7.12 (m, 2H), 6.98 (d, *J* = 8.2 Hz, 1H), 4.30 (s, 4H), 4.22 (m, 2H), 3.37 (m, 2H), 3.28 (m, 2H); ^13^C NMR (101 MHz, DMSO) δ 151.8, 146.7, 144.7, 144.4, 143.3, 141.4, 140.1, 138.1, 131.2, 130.7, 127.6, 126.1, 122.2, 119.2, 118.3, 117.7, 112.8, 65.1, 65.0, 47.2, 42.0, 29.2. UPLC/MS (ESI): *m/z* = 413.3 [M+H]^+^.


***N-((3-(2,3-dihydrobenzo[b][1,4]dioxin-6-yl)-1-phenyl-1H-pyrazol-4-yl)methyl)-2-(pyridin-2-yl)ethan-1-amine hydrochloride* (21).**


Brown solid, 20% yield. ^1^H NMR (400 MHz, DMSO) δ 9.72 (s, 2H, N*H*_2_), 8.93 (s, 1H), 8.70 (d, *J* = 7.2 Hz, 1H), 8.23 (t, *J* = 7.2 Hz, 1H), 7.81 (d, *J* = 8.0 Hz, 2H), 7.75 (d, *J* = 7.2 Hz, 1H), 7.68 (t, *J* = 7.2 Hz, 1H), 7.56 (t, *J* = 8.0 Hz, 2H), 7.37 (t, *J* = 8.0 Hz, 1H), 7.22–7.13 (m, 2H), 6.98 (d, *J* = 8.2 Hz, 1H), 4.34–4.20 (m, 6H), 3.61–3.34 (m, 4H); ^13^C NMR (101 MHz, DMSO) δ 154.5, 151.8, 144.7, 144.4, 140.1, 131.2, 130.7, 127.7, 127.3, 126.1, 125.4, 122.2, 119.3, 118.3, 117.6, 112.8, 65.2, 65.1, 45.9, 42.1, 31.2. UPLC/MS (ESI): *m/z* = 413.3 [M+H]^+^.


***4-(4-(((3-(2,3-dihydrobenzo[b][1,4]dioxin-6-yl)-1-phenyl-1H-pyrazol-4-yl)methyl)amino)phenoxy)-N-methylpicolinamide* (22).**


White solid, 26% yield. ^1^H NMR (400 MHz, DMSO) δ 8.76 (q, *J* = 4.7 Hz, 1H, N*H*), 8.59 (s, 1H), 8.46 (d, *J* = 5.6 Hz, 1H), 7.87 (d, *J* = 7.7 Hz, 2H), 7.52 (t, *J* = 7.7 Hz, 2H), 7.35 (d, *J* = 2.6 Hz, 1H), 7.34–7.28 (m, 3H), 7.10 (dd, *J* = 5.6, 2.6 Hz, 1H), 6.99 (d, *J* = 8.9 Hz, 2H), 6.95 (d, *J* = 9.0 Hz, 1H), 6.78 (d, *J* = 8.9 Hz, 2H), 6.12 (t, *J* = 4.8 Hz, 1H, N*H*), 4.28 (s, 4H), 4.22 (d, *J* = 4.8 Hz, 2H), 2.78 (d, *J* = 4.7 Hz, 3H); ^13^C NMR (101 MHz, DMSO) δ 167.7, 164.8, 153.2, 151.1, 151.1, 147.8, 144.4, 144.3, 144.2, 140.4, 130.5, 129.9, 127.1, 127.0, 122.6, 121.4, 119.2, 118.9, 118.2, 116.9, 114.7, 114.3, 109.3, 65.1, 65.0, 39.5, 26.9. UPLC/MS (ESI): *m/z* = 534.2 [M+H]^+^.


***3-(2,3-dihydrobenzo[b][1,4]dioxin-6-yl)-1-phenyl-4-(pyrrolidin-1-ylmethyl)-1H-pyrazole oxalate* (23).**


Yellow solid, 35% yield. ^1^H NMR (400 MHz, DMSO) δ 8.71 (s, 1H), 7.87 (d, *J* = 8.0 Hz, 2H), 7.54 (t, *J* = 8.0 Hz, 2H), 7.36 (t, *J* = 8.0 Hz, 1H), 7.26–7.21 (m, 1H), 7.19 (dd, *J* = 8.3, 2.1 Hz, 1H), 6.98 (d, *J* = 8.3 Hz, 1H), 4.30 (m, 6H), 3.16 (m, 4H), 1.88 (m, 4H); ^13^C NMR (101 MHz, DMSO) δ 165.1, 152.1, 144.7, 144.4, 140.1, 131.1, 130.6, 127.6, 126.3, 122.1, 119.3, 118.3, 117.7, 65.2, 65.1, 53.7, 48.6, 23.6. UPLC/MS (ESI): *m/z* = 362.3 [M+H]^+^.


***1-((3-(2,3-dihydrobenzo[b][1,4]dioxin-6-yl)-1-phenyl-1H-pyrazol-4-yl)methyl)-4-methylpiperazine* (24).**


Yellow oil, 60% yield. ^1^H NMR (400 MHz, CDCl_3_) δ 7.87 (s, 1H), 7.73 (d, *J* = 7.6 Hz, 2H), 7.54 (d, *J* = 2.0 Hz, 1H), 7.47–7.38 (m, 3H), 7.23 (t, *J* = 7.6 Hz, 1H), 6.91 (d, *J* = 8.4 Hz, 1H), 4.27 (s, 4H), 3.48 (s, 2H), 2.90–2.37 (m, 8H), 2.30 (s, 3H); ^13^C NMR (101 MHz, CDCl_3_) δ 152.1, 143.6, 143.5, 140.1, 129.4, 128.1, 127.0, 126.1, 121.6, 118.7, 117.5, 117.2, 64.6, 64.4, 55.2, 52.7, 52.4, 46.0. UPLC/MS (ESI): *m/z* = 391.3 [M+H]^+^.


***(1r,3r)-N-((3-(2,3-dihydrobenzo[b][1,4]dioxin-6-yl)-1-phenyl-1H-pyrazol-4-yl)methyl)adamantan-2-amine hydrochloride* (25).**


White solid, 13% yield. ^1^H NMR (400 MHz, DMSO) δ 8.46 (s, 1H), 7.85 (d, *J* = 7.8 Hz, 2H), 7.56–7.46 (m, 3H), 7.40 (dd, *J* = 8.3, 2.1 Hz, 1H), 7.28 (t, *J* = 7.8 Hz, 1H), 6.91 (d, *J* = 8.3 Hz, 1H), 4.27 (s, 4H), 3.68 (m, 2H), 2.77 (m, 1H), 2.12 (m, 2H), 1.87 (m, 2H), 1.84–1.72 (m, 5H), 1.68 (m, 4H), 1.42 (m, 2H). ^13^C NMR (101 MHz, DMSO) δ 151.1, 144.2, 144.2, 140.5, 130.4, 129.5, 127.4, 126.7, 121.5, 121.3, 118.8, 117.9, 117.3, 65.1, 65.0, 62.7, 38.5, 37.9, 32.4, 31.9, 28.1. UPLC/MS (ESI): *m/z* = 442.4 [M+H]^+^.

***N-((3-(2,3-dihydrobenzo[b]****[1,4]****dioxin-6-yl)-1-phenyl-1H-pyrazol-4-yl)methyl)-2-(pyrrolidin-1-yl)ethan-1-amine dihydrochloride* (26)** [21].

White solid, 21% yield. ^1^H NMR (400 MHz, DMSO) δ 11.03 (s, 1H, N*H*), 9.79 (s, 2H, N*H*_2_), 8.90 (s, 1H), 7.79 (d, *J* = 8.0 Hz, 2H), 7.56 (t, *J* = 8.0 Hz, 2H), 7.37 (t, *J* = 8.0 Hz, 1H), 7.19 (m, 2H), 6.99 (d, *J* = 8.1 Hz, 1H), 4.30 (m, 6H), 3.76–3.41 (m, 6H), 3.05 (m, 2H), 2.07–1.81 (m, 4H); ^13^C NMR (101 MHz, DMSO) δ 151.7, 144.7, 144.4, 140.1, 130.8, 127.8, 126.0, 122.1, 119.3, 118.3, 117.6, 112.8, 65.2, 65.1, 54.2, 50.5, 43.4, 42.4, 23.5. UPLC/MS (ESI): *m/z* = 405.4 [M+H]^+^.


***N-((3-(2,3-dihydrobenzo[b][1,4]dioxin-6-yl)-1-phenyl-1H-pyrazol-4-yl)methyl)-2-(4-methylpiperazin-1-yl)ethan-1-amine trihydrochloride* (27).**


Orange solid, 28% yield. ^1^H NMR (400 MHz, DMSO) δ 11.53 (m, 1H, N*H*), 9.56 (m, 2H, N*H*), 8.70 (s, 1H), 7.49 (d, *J* = 8.0 Hz, 2H), 7.26 (t, *J* = 8.0 Hz, 2H), 7.07 (t, *J* = 8.0 Hz, 1H), 6.95–6.81 (m, 2H), 6.68 (d, *J* = 8.2 Hz, 1H), 4.06–3.86 (m, 7H), 3.51–2.63 (m, 13H), 2.20 (t, *J* = 1.9 Hz, 2H); ^13^C NMR (101 MHz, DMSO) δ 151.3, 144.4, 144.3, 140.4, 130.5, 130.0, 127.1, 127.0, 121.7, 119.0, 118.1, 117.7, 65.1, 65.0, 56.8, 55.3, 53.2, 46.3, 45.7, 43.6. UPLC/MS (ESI): *m/z* = 434.4 [M+H]^+^.


***N-((3-(2,3-dihydrobenzo[b][1,4]dioxin-6-yl)-1-phenyl-1H-pyrazol-4-yl)methyl)-2-morpholinoethan-1-amine dihydrochloride* (28).**


White solid, 40% yield. ^1^H NMR (400 MHz, DMSO) δ 11.36 (s, 1H, N*H*), 9.91 (s, 2H, N*H*_2_), 8.94 (s, 1H), 7.79 (d, *J* = 8.0 Hz, 2H), 7.56 (t, *J* = 8.0 Hz, 2H), 7.37 (t, *J* = 8.0 Hz, 1H), 7.22–7.12 (m, 2H), 6.98 (d, *J* = 8.1 Hz, 1H), 4.33–4.25 (m, 6H), 3.99 (m, 2H), 3.81 (m, 2H), 3.55 (m, 6H), 3.16 (m, 2H) ^13^C NMR (101 MHz, DMSO) δ 151.6, 144.7, 144.4, 140.1, 130.8, 130.7, 127.7, 126.0, 122.1, 119.2, 118.3, 117.6, 112.7, 65.1, 65.0, 64.2, 52.9, 52.5, 42.3, 41.4. UPLC/MS (ESI): *m/z* = 421.3 [M+H]^+^.

***N^1^-((3-(2,3-dihydrobenzo[b]****[1,4]****dioxin-6-yl)-1-phenyl-1H-pyrazol-4-yl)methyl)-N^3^,N^3^-dimethylpropane-1,3-diamine dihydrochloride* (29)** [21].

Orange solid, 77% yield. ^1^H NMR (400 MHz, DMSO) δ 10.47 (brs, 1H, N*H*), 9.52 (brs, 2H, N*H_2_*), 8.93 (s, 1H), 7.81 (d, *J* = 7.8 Hz, 2H), 7.56 (t, *J* = 7.8 Hz, 2H), 7.37 (t, *J* = 7.8 Hz, 1H), 7.24–7.09 (m, 2H), 6.99 (d, *J* = 8.2 Hz, 1H), 4.30 (s, 4H), 4.20 (t, *J* = 5.6 Hz, 2H), 3.16 (m, 2H), 3.08 (m, 2H), 2.74 (m, 6H), 2.10 (p, *J* = 7.5 Hz, 2H); ^13^C NMR (101 MHz, DMSO) δ 151.8, 144.7, 144.4, 140.1, 131.2, 130.7, 127.7, 126.1, 122.2, 119.3, 118.3, 117.7, 112.8, 65.2, 65.1, 54.5, 44.6, 42.9, 41.9, 21.6. UPLC/MS (ESI): *m/z* = 393.3 [M+H]^+^.


***3-(2,3-dihydrobenzo[b][1,4]dioxin-6-yl)-N,1-diphenyl-1H-pyrazole-4-carboxamide* (30).**


White solid, Yield 45%. ^1^H NMR (400 MHz, CDCl_3_) δ 8.59 (s, 1H), 7.76 (d, *J* = 7.8 Hz, 2H), 7.66 (s, 1H, N*H*), 7.48 (t, *J* = 7.8 Hz, 2H), 7.41–7.27 (m, 6H), 7.21 (dd, *J* = 8.2, 2.1 Hz, 1H), 7.09 (t, *J* = 7.8 Hz, 1H), 7.04 (d, *J* = 8.2 Hz, 1H), 4.34 (q, *J* = 5.3 Hz, 4H); ^13^C NMR (101 MHz, CDCl_3_) δ 160.7, 150.4, 145.0, 144.3, 139.4, 137.9, 131.6, 129.7, 129.1, 127.5, 125.2, 124.3, 122.7, 119.8, 119.5, 118.8, 118.4, 118.1, 64.7, 64.5. UPLC/MS (ESI): *m/z* = 398.3 [M+H]^+^.


***3-(2,3-dihydrobenzo[b][1,4]dioxin-6-yl)-N-(4-fluorophenyl)-1-phenyl-1H-pyrazole-4-carboxamide* (31).**


Pale yellow solid, 67% yield. ^1^H NMR (400 MHz, DMSO) δ 10.30 (s, 1H, N*H*), 9.04 (s, 1H), 7.93 (d, *J* = 7.8 Hz, 2H), 7.72 (m, 2H), 7.57 (t, *J* = 7.8 Hz, 2H), 7.45–7.31 (m, 3H), 7.20 (m, 2H), 6.91 (d, *J* = 8.4 Hz, 1H), 4.27 (s, 4H); ^13^C NMR (101 MHz, DMSO) δ 162.5, 159.1 (d, ^1^*J*_C-F_ = 238 Hz), 151.3, 144.7, 143.9 139.9, 136.5 (d, ^4^*J*_C-F_ = 2 Hz), 131.4, 130.7, 127.9, 126.3, 122.4 (d, ^3^*J*_C-F_ = 8 Hz), 122.1, 119.5, 118.5, 117.79, 117.74, 116.3 (d, ^2^*J*_C-F_ 22 = Hz), 65.2, 65.0. UPLC/MS (ESI): *m/z* = 416.3 [M+H]^+^.


***N-benzyl-3-(2,3-dihydrobenzo[b][1,4]dioxin-6-yl)-1-phenyl-1H-pyrazole-4-carboxamide* (32).**


White solid, 84% yield. ^1^H NMR (400 MHz, DMSO) δ 8.91 (s, 1H), 8.72 (t, *J* = 6.0 Hz, 1H, N*H*), 7.88 (d, *J* = 7.8 Hz, 2H), 7.54 (t, *J* = 7.8 Hz, 2H), 7.42 (d, *J* = 2.0 Hz, 1H), 7.38–7.32 (m, 6H), 7.26 (m, 1H), 6.87 (d, *J* = 8.4 Hz, 1H), 4.44 (d, *J* = 6.0 Hz, 2H), 4.27 (s, 4H); ^13^C NMR (101 MHz, DMSO) δ 163.8, 151.2, 144.6, 143.8, 140.3, 140.0, 130.9, 130.6, 129.2, 128.3, 127.8, 127.7, 126.5, 122.3, 119.5, 118.4, 118.1, 117.5, 65.2, 65.0, 43.3. UPLC/MS (ESI): *m/z* = 412.3 [M+H]^+^.


***3-(2,3-dihydrobenzo[b][1,4]dioxin-6-yl)-N-(4-fluorobenzyl)-1-phenyl-1H-pyrazole-4-carboxamide* (33).**


Yellow solid, 65% yield. ^1^H NMR (400 MHz, DMSO) δ 8.90 (s, 1H), 8.73 (t, *J* = 6.0 Hz, 1H, N*H*), 7.88 (d, *J* = 7.4 Hz, 2H), 7.54 (t, *J* = 7.4 Hz, 2H), 7.45–7.31 (m, 5H), 7.17 (m, 2H), 6.87 (d, *J* = 8.4 Hz, 1H), 4.42 (d, *J* = 6.0 Hz, 2H), 4.27 (s, 4H); ^13^C NMR (101 MHz, DMSO) δ 163.8, 162.1 (d, ^1^*J*_C-F_ = 240 Hz), 151.2, 144.6, 143.8, 140.0, 136.6 (d, ^4^*J*_C-F_ = 3 Hz), 130.9, 130.6, 130.3 (d, ^3^*J*_C-F_ = 8 Hz), 127.8, 126.4, 122.3, 119.5, 118.3, 118.0, 117.5, 115.9 (d, ^2^*J*_C-F_ = 21 Hz), 65.2, 65.00, 42.7. UPLC/MS (ESI): *m/z* = 430.3 [M+H]^+^.


***3-(2,3-dihydrobenzo[b][1,4]dioxin-6-yl)-N-phenethyl-1-phenyl-1H-pyrazole-4-carboxamide* (34).**


White solid, 56% yield. ^1^H NMR (400 MHz, DMSO) δ 8.78 (s, 1H), 8.29 (t, *J* = 6.8 Hz, 1H, N*H*), 7.86 (d, *J* = 7.8 Hz, 2H), 7.55 (t, *J* = 7.8 Hz, 2H), 7.40 (d, *J* = 2.0 Hz, 1H), 7.36 (t, *J* = 7.8 Hz, 1H), 7.33–7.18 (m, 6H), 6.87 (d, *J* = 8.4 Hz, 1H), 4.28 (s, 4H), 3.45 (q, *J* = 6.8 Hz, 2H), 2.83 (t, *J* = 6.8 Hz, 2H); ^13^C NMR (101 MHz, DMSO) δ 163.7, 151.0, 144.6, 143.8, 140.3, 140.0, 130.7, 130.6, 129.6, 129.3, 127.7, 127.1, 126.4, 122.2, 119.4, 118.6, 117.9, 117.6, 65.2, 65.0, 41.4, 36.1. UPLC/MS (ESI): *m/z* = 426.3 [M+H]^+^.


***3-(2,3-dihydrobenzo[b][1,4]dioxin-6-yl)-1-phenyl-N-(2-(pyridin-4-yl)ethyl)-1H-pyrazole-4-carboxamide* (35).**


White solid, 55% yield. ^1^H NMR (400 MHz, DMSO) δ 8.76 (s, 1H), 8.47 (d, *J* = 6.0 Hz, 2H), 8.30 (t, *J* = 6.8 Hz, 1H, N*H*), 7.85 (d, *J* = 8.0 Hz, 2H), 7.54 (t, *J* = 8.0 Hz, 2H), 7.42–7.33 (m, 2H), 7.28 (d, *J* = 6.0 Hz, 2H), 7.25 (dd, *J* = 8.4, 2.1 Hz, 1H), 6.86 (d, *J* = 8.4 Hz, 1H), 4.28 (s, 4H), 3.50 (q, *J* = 6.8 Hz, 2H), 2.85 (t, *J* = 6.8 Hz, 2H); ^13^C NMR (101 MHz, DMSO) δ 163.8, 151.0, 150.4, 149.3, 144.6, 143.8, 139.9, 130.7, 130.6, 127.8, 126.4, 125.2, 122.2, 119.4, 118.5, 117.9, 117.6, 65.1, 65.0, 40.2, 35.2. UPLC/MS (ESI): *m/z* = 427.3 [M+H]^+^.


***3-(2,3-dihydrobenzo[b][1,4]dioxin-6-yl)-1-phenyl-N-(2-(pyridin-3-yl)ethyl)-1H-pyrazole-4-carboxamide* (36).**


White solid, 45% yield, ^1^H NMR (400 MHz, DMSO) δ 8.77 (s, 1H), 8.47 (d, *J* = 2.3 Hz, 1H), 8.43 (dd, *J* = 4.8, 2.3 Hz, 1H), 8.33 (t, *J* = 6.7 Hz, 1H, N*H*), 7.86 (d, *J* = 7.8 Hz, 2H), 7.69 (dt, *J* = 7.8, 2.3 Hz, 1H), 7.55 (t, *J* = 7.8 Hz, 2H), 7.42–7.29 (m, 3H), 7.26 (dd, *J* = 8.5, 2.1 Hz, 1H), 6.86 (d, *J* = 8.5 Hz, 1H), 4.27 (s, 4H), 3.48 (q, *J* = 6.7 Hz, 2H), 2.84 (t, *J* = 6.7 Hz, 2H); ^13^C NMR (101 MHz, DMSO) δ 163.8, 151.0, 150.8, 148.4, 144.6, 143.8, 139.9, 137.2, 135.8, 130.7, 130.6, 127.8, 126.4, 124.4, 122.2, 119.4, 118.5, 117.9, 117.6, 65.2, 65.0, 41.0, 33.1. UPLC/MS (ESI): *m/z* = 427.3 [M+H]^+^.


***3-(2,3-dihydrobenzo[b][1,4]dioxin-6-yl)-1-phenyl-N-(2-(pyridin-2-yl)ethyl)-1H-pyrazole-4-carboxamide* (37).**


Pale yellow solid, 56% yield. ^1^H NMR (400 MHz, DMSO) δ 8.79 (s, 1H), 8.50 (d, *J* = 4.7 Hz, 1H), 8.29 (t, *J* = 6.9 Hz, 1H, N*H*), 7.86 (d, *J* = 7.9 Hz, 2H), 7.71 (td, *J* = 7.6, 1.9 Hz, 1H), 7.54 (t, *J* = 7.9 Hz, 2H), 7.41–7.33 (m, 2H), 7.30 (m, 2H), 7.26–7.19 (m, 1H), 6.87 (d, *J* = 8.4 Hz, 1H), 4.27 (s, 4H), 3.58 (q, *J* = 6.9 Hz, 2H), 2.98 (t, *J* = 6.9 Hz, 2H); ^13^C NMR (101 MHz, DMSO) δ 163.7, 160.0, 151.0, 150.0, 144.6, 143.8, 140.0, 137.4, 130.8, 130.6, 127.7, 126.4, 124.1, 122.5, 122.2, 119.4, 118.6, 117.9, 117.6, 65.2, 65.0, 39.8, 38.2. UPLC/MS (ESI): *m/z* = 427.3 [M+H]^+^.


***4-(4-(3-(2,3-dihydrobenzo[b][1,4]dioxin-6-yl)-1-phenyl-1H-pyrazole-4-carboxamido)phenoxy)-N-methylpicolinamide* (38).**


Pale yellow solid, 69% yield. ^1^H NMR (400 MHz, DMSO) δ 10.41 (s, 1H, N*H*), 9.07 (s, 1H), 8.79 (q, *J* = 5.0 Hz, 1H, N*H*), 8.52 (d, *J* = 5.6 Hz, 1H), 7.94 (d, *J* = 9.0 Hz, 2H), 7.84 (d, *J* = 7.8 Hz, 2H), 7.58 (t, *J* = 7.8 Hz, 2H), 7.47–7.33 (m, 4H), 7.24 (d, *J* = 9.0 Hz, 2H), 7.17 (dd, *J* = 5.6, 2.6 Hz, 1H), 6.93 (d, *J* = 8.4 Hz, 1H), 4.28 (s, 4H), 2.79 (d, *J* = 5.0 Hz, 3H); ^13^C NMR (101 MHz, DMSO) δ 166.8, 164.7, 162.6, 153.4, 151.4, 151.3, 149.6, 144.7, 143.9, 139.9, 137.9, 131.5, 130.7, 127.9, 126.2, 122.4, 122.3, 122.1, 119.5, 118.5, 117.8, 117.8, 115.0, 109.7, 65.2, 65.0, 27.4. UPLC/MS (ESI): *m/z* = 548.2 [M+H]^+^.


***(3-(2,3-dihydrobenzo[b][1,4]dioxin-6-yl)-1-phenyl-1H-pyrazol-4-yl)(pyrrolidin-1-yl)methanone* (39).**


White solid, 65% yield. ^1^H NMR (400 MHz, DMSO) δ 8.78 (s, 1H), 7.91 (d, *J* = 8.1 Hz, 2H), 7.53 (t, *J* = 8.1 Hz, 2H), 7.35 (t, *J* = 8.1 Hz, 1H), 7.27 (d, *J* = 2.1 Hz, 1H), 7.24 (dd, *J* = 8.4, 2.1 Hz, 1H), 6.92 (d, *J* = 8.4 Hz, 1H), 4.28 (s, 4H), 3.47 (t, *J* = 6.8 Hz, 2H), 3.27 (t, *J* = 6.8 Hz, 2H), 1.81 (m, 4H); ^13^C NMR (101 MHz, DMSO) δ 163.9, 149. 5, 144.6, 144.2, 140.1, 130.5, 129.3, 127.5, 126.5, 120.9, 119.4, 118.7, 118.1, 116.4, 65.1, 65.0, 49.0, 46.5, 26.4, 25.0. UPLC/MS (ESI): *m/z* = 376.3 [M+H]^+^.


***(3-(2,3-dihydrobenzo[b][1,4]dioxin-6-yl)-1-phenyl-1H-pyrazol-4-yl)(morpholino)methanone* (40).**


White solid, 18% yield. ^1^H NMR (400 MHz, DMSO) δ 8.78 (s, 1H), 7.94 (d, *J* = 7.8 Hz, 2H), 7.57 (t, *J* = 7.8 Hz, 2H), 7.39 (t, *J* = 7.8 Hz, 1H), 7.28–7.16 (m, 2H), 7.00 (d, *J* = 8.2 Hz, 1H), 4.33 (s, 4H), 3.65 (m, 4H), 3.31 (m, 4H); ^13^C NMR (101 MHz, DMSO) δ 164.5, 149.6, 144.8, 144.3, 140.0, 130.5, 129.7, 127.6, 126.1, 121.0, 119.4, 118.3, 116.7, 116.5, 66.8, 65.2, 65.1. UPLC/MS (ESI): *m/z* = 392.3 [M+H]^+^.


***N-((1r,3r,5r,7r)-adamantan-2-yl)-3-(2,3-dihydrobenzo[b][1,4]dioxin-6-yl)-1-phenyl-1H-pyrazole-4-carboxamide* (41).**


White solid, 21% yield. ^1^H NMR (400 MHz, DMSO) δ 8.86 (s, 1H), 7.92 (d, *J* = 8.1 Hz, 2H), 7.73 (d, *J* = 7.2 Hz, 1H, N*H*), 7.53 (t, *J* = 8.1 Hz, 2H), 7.40–7.28 (m, 3H), 6.91 (d, *J* = 8.3 Hz, 1H), 4.27 (s, 4H), 4.01 (d, *J* = 7.2 Hz, 1H), 2.00–1.87 (m, 4H), 1.87–1.73 (m, 6H), 1.70 (m, 2H), 1.49 (d, *J* = 12.5 Hz, 2H); ^13^C NMR (101 MHz, DMSO) δ 163.3, 150.8, 144.6, 143.9, 140.1, 131.1, 130.5, 127.6, 126.5, 122.2, 119.5, 119.0, 117.9, 117.8, 65.2, 65.0, 54.5, 38.1, 37.8, 32.3, 27.7. UPLC/MS (ESI): *m/z* = 456.4 [M+H]^+^.


***3-(2,3-dihydrobenzo[b][1,4]dioxin-6-yl)-1-phenyl-N-(2-(pyrrolidin-1-yl)ethyl)-1H-pyrazole-4-carboxamide oxalate* (42).**


Brown solid, 33% yield. ^1^H NMR (400 MHz, MeOD) δ 8.68 (s, 1H), 7.83 (d, *J* = 7.9 Hz, 2H), 7.53 (t, *J* = 7.9 Hz, 2H), 7.49–7.20 (m, 3H), 6.89 (d, *J* = 8.4 Hz, 1H), 4.28 (s, 4H), 3.72 (m, 4H), 3.41 (m, 2H), 3.14 (m, 2H), 2.05 (m, 4H); ^13^C NMR (101 MHz, MeOD) δ 166.8, 164.5, 153.1, 145.8, 144.7, 140.7, 131.5, 130.7, 128.5, 126.7, 123.1, 120.4, 118.9, 118.0, 117.6, 65.8, 65.65, 55.8, 55.6, 37.2, 24.0. UPLC/MS (ESI): *m/z* = 419.4 [M+H]^+^.


***3-(2,3-dihydrobenzo[b][1,4]dioxin-6-yl)-N-(2-(4-methylpiperazin-1-yl)ethyl)-1-phenyl-1H-pyrazole-4-carboxamide* (43).**


White solid, 33% yield. ^1^H NMR (400 MHz, DMSO) δ 8.80 (s, 1H), 7.97 (t, *J* = 5.7 Hz, 1H, N*H*), 7.88 (t, *J* = 8.1 Hz, 2H), 7.54 (t, *J* = 8.1 Hz, 2H), 7.42–7.30 (m, 3H), 6.89 (d, *J* = 8.4 Hz, 1H), 4.28 (s, 4H), 3.32 (m, 2H), 2.47–2.14 (m, 13H); ^13^C NMR (101 MHz, DMSO) δ 163.6, 150.9, 144.6, 143.8, 140.0, 130.9, 130.6, 127.7, 126.4, 122.2, 119.4, 118.7, 117.9, 117.6, 65.1, 65.0, 57.8, 55.6, 53.6, 46.7, 37.4. UPLC/MS (ESI): *m/z* = 448.3 [M+H]^+^.


***3-(2,3-dihydrobenzo[b][1,4]dioxin-6-yl)-N-(2-morpholinoethyl)-1-phenyl-1H-pyrazole-4-carboxamide hydrochloride* (44).**


Yellow solid, 26% yield. ^1^H NMR (400 MHz, DMSO) δ 10.89 (brs, 1H, N*H*), 9.20 (s, 1H), 8.67 (t, *J* = 6.0 Hz, 1H, N*H*), 7.87 (d, *J* = 7.9 Hz, 2H), 7.71–7.19 (m, 5H), 6.89 (d, *J* = 8.5 Hz, 1H), 4.27 (s, 4H), 4.00–3.79 (m, 4H), 3.65 (m, 2H), 3.53 (m, 2H), 3.28 (m, 2H), 3.13 (m, 2H); ^13^C NMR (101 MHz, DMSO) δ 164.0, 151.6, 144.6, 143.7, 139.9, 131.7, 130.7, 127.9, 126.4, 122.6, 119.4, 118.4, 117.7, 117.5, 65.2, 65.0, 64.1, 56.4, 52.1, 42.4, 34.3. UPLC/MS (ESI) *m/z* = 435.3 [M+H]^+^.


***3-(2,3-dihydrobenzo[b][1,4]dioxin-6-yl)-4-(((4-fluorobenzyl)oxy)methyl)-1-phenyl-1H-pyrazole* (45).**


Pale yellow solid, 15% yield. ^1^H NMR (400 MHz, CDCl_3_) δ 7.80 (s, 1H), 7.60–7.53 (m, 2H), 7.34–7.24 (m, 3H), 7.24–7.15 (m, 3H), 7.11 (t, *J* = 7.3 Hz, 1H), 6.88 (m, 2H), 6.77 (d, *J* = 8.3 Hz, 1H), 4.42 (s, 2H), 4.40 (s, 2H), 4.13 (s, 4H); ^13^C NMR (101 MHz, CDCl_3_) δ 162.5 (d, ^1^*J*_C-F_ 250 Hz), 151.8, 143.8, 143.6, 140.0, 133.8 (d, ^4^*J*_C-F_ 3 Hz), 129.9 (d, ^3^*J*_C-F_ 8 Hz), 129.5, 128.4, 126.6, 126.4, 121.3, 118.9, 117.4, 117.0, 115.4 (d, ^2^*J*_C-F_ 21 Hz), 71.5, 64.6, 64.4, 63.0. UPLC/MS (ESI) *m/z* = 417.3 [M+H]^+^.


***4-(((3-(2,3-dihydrobenzo[b][1,4]dioxin-6-yl)-1-phenyl-1H-pyrazol-4-yl)methoxy)methyl)-1-phenyl-1H-1,2,3-triazole* (46).**


Pale yellow solid, 65% yield. ^1^H NMR (400 MHz, DMSO) δ 8.88 (s, 1H), 8.64 (s, 1H), 7.92 (d, *J* = 7.9 Hz, 2H), 7.88 (d, *J* = 7.5 Hz, 2H), 7.62 (t, *J* = 7.9 Hz, 2H), 7.55–7.46 (m, 3H), 7.36–7.25 (m, 3H), 6.90 (d, *J* = 9.0 Hz, 1H), 4.73 (s, 2H), 4.58 (s, 2H), 4.34–4.08 (m, 4H); ^13^C NMR (101 MHz, DMSO) δ 151.7, 145.8, 144.4, 144.3, 140.3, 137.6, 131.0, 130.9, 130.5, 129.6, 127.2, 126.7, 123.4, 121.4, 121.0, 119.1, 118.1, 117.9, 116.9, 65.1, 65.0, 63.2, 63.0. UPLC/MS (ESI) *m/z* = 466.3 [M+H]^+^.


***4-(((3-(2,3-dihydrobenzo[b][1,4]dioxin-6-yl)-1-phenyl-1H-pyrazol-4-yl)methoxy)methyl)-1-(4-fluorophenyl)-1H-1,2,3-triazole* (47).**


Pale pink solid. 21% yield. ^1^H NMR (400 MHz, CDCl_3_) δ 8.06 (s, 1H), 7.99 (s, 1H), 7.80–7.68 (m, 4H), 7.51–7.42 (m, 3H), 7.39 (dd, *J* = 8.4, 2.1 Hz, 1H), 7.35–7.17 (m, 3H), 6.95 (d, *J* = 8.4 Hz, 1H), 4.87 (s, 2H), 4.67 (s, 2H), 4.31–4.17 (m, 4H); ^13^C NMR (101 MHz, CDCl_3_) δ 162.6 (d, ^1^*J*_C-F_ = 248 Hz), 152.0, 146.2, 143.9, 143.6, 140.0, 133.5, 129.6, 128.9, 126.6, 126.5, 122.7 (d, ^3^*J*_C-F_ = 9 Hz), 121.4, 121.3, 119.1, 117.6, 117.1, 117.1, 116.8 (d, ^2^*J*_C-F_ = 23 Hz), 64.6, 64.5, 63.3, 63.2. UPLC/MS (ESI) *m/z* = 484.3 [M+H]^+^.


***1-benzyl-4-(((3-(2,3-dihydrobenzo[b][1,4]dioxin-6-yl)-1-phenyl-1H-pyrazol-4-yl)methoxy)methyl)-1H-1,2,3-triazole* (48).**


White solid, 56% yield. ^1^H NMR (400 MHz, DMSO) δ 8.59 (s, 1H), 8.21 (s, 1H), 7.87 (d, *J* = 8.0 Hz, 2H), 7.50 (t, *J* = 8.0 Hz, 2H), 7.26–7.40 (m, 8H), 6.90 (d, *J* = 8.3 Hz, 1H), 5.61 (s, 2H), 4.63 (s, 2H), 4.51 (s, 2H), 4.27 (s, 4H); ^13^C NMR (101 MHz, DMSO) δ 151.7, 145.0, 144.5, 144.3, 140.3, 137.0, 130.9, 130.5, 129.7, 129.1, 128.8, 127.1, 126.8, 125.2, 121.4, 119.1, 118.2, 117.9, 116.9, 65.1, 65.1, 63.3, 63.0, 53.7. UPLC/MS (ESI) *m/z* = 480.3 [M+H]^+^.


***4-benzyl-3-(2,3-dihydrobenzo[b][1,4]dioxin-6-yl)-1-phenyl-1H-pyrazole* (49).**


Pale yellow solid, 5% yield. ^1^H NMR (400 MHz, CDCl_3_) δ 7.68 (d, *J* = 8.0 Hz, 2H), 7.55 (s, 1H), 7.41 (t, *J* 8.0 Hz, 2H), 7.36–7.29 (m, 2H), 7.29–7.19 (m, 6H), 6.91 (d, *J* = 8.4 Hz, 1H), 4.29 (s, 4H), 4.03 (s, 2H);^13^C NMR (101 MHz, CDCl_3_) δ 151.2, 143.6, 140.6, 140.2, 129.4, 128.8, 128.7, 127.4, 127.1, 126.4, 126.1, 121.3, 120.4, 118.7, 117.4, 117.0, 64.6, 64.5, 31.2. UPLC/MS (ESI) *m/z* = 369.3 [M+H]^+^.


***4-(4-chlorobenzyl)-3-(2,3-dihydrobenzo[b][1,4]dioxin-6-yl)-1-phenyl-1H-pyrazole* (50).**


White solid, 18% yield. ^1^H NMR (400 MHz, CDCl_3_) δ 7.69 (d, *J* = 8.0 Hz, 2H), 7.55 (s, 1H), 7.42 (t, *J* = 8.0 Hz, 2H), 7.32–7.22 (m, 4H), 7.22–7.13 (m, 3H), 6.91 (d, *J* = 8.3 Hz, 1H), 4.29 (s, 4H), 4.00 (s, 2H); ^13^C NMR (101 MHz, CDCl_3_) δ 151.1, 143.7, 143.7, 140.1, 139.1, 132.2, 130.1, 129.5, 128.9, 127.4, 126.9, 126.3, 121.3, 119.9, 118.8, 117.5, 117.0, 64.6, 64.5, 30.6. UPLC/MS (ESI) *m/z* = 403.3 [M+H]^+^.


***3-(2,3-dihydrobenzo[b][1,4]dioxin-6-yl)-4-(4-fluorobenzyl)-1-phenyl-1H-pyrazole* (51).**


White solid, 50% yield. ^1^H NMR (400 MHz, DMSO) δ 8.31 (s, 1H), 7.84 (d, *J* = 7.9 Hz, 2H), 7.48 (t, *J* = 7.9 Hz, 2H), 7.33–7.20 (m, 3H), 7.20–7.06 (m, 4H), 6.89 (d, *J* = 8.3 Hz, 1H), 4.25 (s, 4H), 4.01 (s, 2H); ^13^C NMR (101 MHz, DMSO) δ 161.6 (d, ^1^*J*_C-F_ = 240 Hz), 150.9, 144.2, 140.4, 137.4 (d, ^4^*J*_C-F_ = 3 Hz), 130.9 (d, ^3^*J*_C-F_ = 8 Hz), 130.4, 129.5, 127.2, 126.9, 121.3, 120.1, 118.9, 118.1, 116.8, 116.1 (d, ^2^*J*_C-F_ = 22 Hz), 65.1, 65.0, 30.0. UPLC/MS (ESI) *m/z* = 387.3 [M+H]^+^.


***3-(2,3-dihydrobenzo[b][1,4]dioxin-6-yl)-4-(fluoromethyl)-1-phenyl-1H-pyrazole* (52).**


White solid, 50% yield, ^1^H NMR (400 MHz, CDCl_3_) δ 8.04 (d, *J* = 3.2 Hz, 1H), 7.72 (d, *J* = 7.4 Hz, 2H), 7.49–7.39 (m, 2H), 7.34 (d, *J* = 2.1 Hz, 1H), 7.32–7.25 (m, 2H), 6.94 (d, *J* = 8.3 Hz, 1H), 5.40 (d, *J_CH2-F_* = 49.5 Hz, 2H), 4.28 (s, 4H); ^13^C NMR (101 MHz, CDCl_3_) δ 152.6 (d, ^3^*J* = 2.1 Hz), 144.2, 143.8, 139.9, 129.6, 129.4, 126.9, 126.0, 121.4 (d, ^3^*J_C-F_* = 2.2 Hz), 119.3, 117.7, 117.1 (d, ^4^*J_C-F_* = 2.2 Hz), 116.0 (d, ^2^*J_C-F_* = 21.1 Hz), 75.49 (d, ^1^*J_C-F_* = 161.9 Hz), 64.67, 64.51. UPLC/MS (ESI) *m/z* = 311.1 [M+H]^+^.


***4-(difluoromethyl)-3-(2,3-dihydrobenzo[b][1,4]dioxin-6-yl)-1-phenyl-1H-pyrazole* (53).**


Pale yellow solid, 76% yield. ^1^H NMR (400 MHz, CDCl_3_) δ 8.20 (s, 1H), 7.75 (d, *J* = 7.5 Hz, 2H), 7.48 (t, *J* = 7.5 Hz, 2H), 7.34 (t, *J* = 7.5 Hz, 1H), 7.30 (d, *J* = 2.1 Hz, 1H), 7.24 (dd, *J* = 8.4, 2.1 Hz, 1H), 6.96 (d, *J* = 8.4 Hz, 1H), 6.80 (t, *J* = 55.4 Hz, 1H), 4.31 (s, 4H); ^13^C NMR (101 MHz, CDCl_3_) δ 150.8 (t, ^3^*J*_C-F_ = 5.2 Hz), 144.4, 143.9, 139.6, 129.7, 127.4 (t, ^3^*J*_C-F_ = 4.8 Hz), 127.3, 125.4, 121.4, 119.5, 117.8, 117.1, 116.1 (t, ^2^*J*_C-F_ = 25.4 Hz), 111.3 (t, ^1^*J*_C-F_ = 233.6 Hz), 64.6, 64.5. UPLC/MS (ESI) *m/z* = 329.1 [M+H]^+^.


***1-((3-(2,3-dihydrobenzo[b][1,4]dioxin-6-yl)-1-phenyl-1H-pyrazol-4-yl)methyl)-3-(4-fluorophenyl)urea* (54).**


A solution of 12 (0.1 g, 1.0 equiv.) was treated with 4-fluorophenyl isocyanate (1.1 equiv.) in acetonitrile (10 mL) and the reaction was continued until the consumption of the starting materials (UPLC monitoring). The reaction was quenched with water, extracted with EtOAc (3 × 15 mL), the combined organic layers were dried over Na_2_SO_4_ and the solvents were evaporated under reduced pressure to obtain a crude that was purified by flash column chromatography to afford the compound **54**.

White solid, 23% yield. ^1^H NMR (400 MHz, DMSO) δ 8.48 (m 2H), 7.86 (d, *J* = 7.9 Hz, 2H), 7.50 (t, *J* = 7.9Hz, 2H), 7.45–7.34 (m, 2H), 7.31 (m, 1H), 7.24 (d, *J* = 6.4 Hz, 2H), 7.06 (m 2H), 6.96 (d, *J* = 8.3 Hz, 1H), 6.48 (brs, 1H, N*H*), 4.40–4.21 (m, 6H). ^13^C NMR (101 MHz, DMSO) δ 157.9 (d, ^1^*J*_C-F_ = 235 Hz), 156.0, 150.5, 144.4 (d, ^4^*J*_C-F_ = 3 Hz), 140.4, 137.7, 130.5, 129.6, 127.1, 127.0, 121.4, 120.2 (d, ^3^*J*_C-F_ = 8 Hz), 119.8, 119.0, 118.2, 116.8, 116.1 (d, ^2^*J*_C-F_ = 22 Hz), 65.1, 65.0, 35.2. UPLC/MS (ESI) *m/z* = 445.3 [M+H]^+^.


**
*1-(3-(benzo[d]*
**
*[1,3]*
***dioxol-5-yl)-1-phenyl-1H-pyrazol-4-yl)-N-(4-fluorobenzyl)methanamine hydrochloride* (55).**


White solid, 36% yield. ^1^H NMR (400 MHz, DMSO) δ 9.65 (brs, 2H, N*H*_2_), 8.84 (s, 1H), 7.81 (d, *J* = 7.8 Hz, 2H), 7.61 (m, 2H), 7.56 (t, *J* = 7.8 Hz, 2H), 7.37 (t, *J* = 7.8 Hz, 1H), 7.27 (m, 2H), 7.19 (d, *J* = 1.7 Hz, 1H), 7.09 (dd, *J* = 7.9, 1.7 Hz, 1H), 7.00 (d, *J* = 7.9 Hz, 1H), 6.09 (s, 2H), 4.23 (t, *J* = 5.2 Hz, 2H), 4.18 (t, *J* = 5.2 Hz, 2H); ^13^C NMR (101 MHz, DMSO) δ 163.3 (d, ^1^*J_C-F_* = 243 Hz), 152.2, 148.5, 148.4, 140.1, 133.6 (d, ^3^*J_C-F_* = 8 Hz), 131.4, 130.7, 129.0 (d, ^4^*J_C-F_* = 3 Hz), 127.7, 126.7, 123.0, 119.3, 116.3 (d, ^2^*J_C-F_* = 21 Hz), 112.8, 109.4, 102.2, 49.7, 41.2. UPLC/MS (ESI) *m/z* = 402.3 [M+H]^+^.


**
*N-((3-(benzo[d]*
**
*[1,3]*
***dioxol-5-yl)-1-phenyl-1H-pyrazol-4-yl)methyl)-2-(4-fluorophenyl)ethan-1-amine hydrochloride* (56).**


White solid, 60% yield. ^1^H NMR (400 MHz, DMSO) δ 9.47 (brs, 2H, N*H*_2_), 8.89 (s, 1H), 7.81 (d, *J* = 8.0 Hz, 2H), 7.56 (t, *J* = 8.0 Hz, 2H), 7.37 (t, *J* = 8.0 Hz, 1H), 7.34–7.23 (m, 3H), 7.23–7.10 (m, 3H), 7.05 (d, *J* = 8.0 Hz, 1H), 6.10 (s, 2H), 4.22 (m, 2H), 3.26–3.15 (m, 2H), 3.11–2.93 (m, 2H); ^13^C NMR (101 MHz, DMSO) δ 162.1 (d, ^1^*J*_C-F_ = 241 Hz), 152.0, 148.6, 148.4, 140.1, 134.4 (d, ^4^*J*_C-F_ = 3 Hz), 131.4 (d, ^3^*J*_C-F_ = 8 Hz), 131.3, 130.7, 127.7, 126.8, 123.1, 119.3, 116.3 (d, ^2^*J*_C-F_ = 21 Hz), 113.1, 109.5, 109.4, 102.2, 48.5, 41.8, 31.5. UPLC/MS (ESI) *m/z* = 416.3 [M+H]^+^.


**
*N-((3-(benzo[d]*
**
*[1,3]*
***dioxol-5-yl)-1-phenyl-1H-pyrazol-4-yl)methyl)-2-(pyridin-4-yl)ethan-1-amine hydrochloride* (57).**


Pale yellow solid, 60% yield. ^1^H NMR (400 MHz, DMSO) δ 9.88 (s, 2H, N*H*_2_), 9.00 (s, 1H), 8.92–8.79 (m, 2H), 7.95 (d, *J* = 4.0 Hz, 2H), 7.86–7.74 (m, 2H), 7.61–7.51 (m, 2H), 7.42–7.33 (m, 1H), 7.27 (d, *J* = 1.7 Hz, 1H), 7.17 (dd, *J* = 8.0, 1.7 Hz, 1H), 7.04 (d, *J* = 8.0 Hz, 1H), 6.10 (s, 2H), 4.23 (m, 2H), 3.37 (m, 4H); ^13^C NMR (101 MHz, DMSO) δ 158.3, 152.0, 148.5, 148.4, 142.5, 140.1, 131.4, 130.7, 128.3, 127.7, 126.8, 123.1, 119.2, 112.9, 109.5, 109.5, 102.2, 46.4, 42.0, 32.2. UPLC/MS (ESI) *m/z* = 399.3 [M+H]^+^.


**
*N-((3-(benzo[d]*
**
*[1,3]*
***dioxol-5-yl)-1-phenyl-1H-pyrazol-4-yl)methyl)-2-(pyridin-3-yl)ethan-1-amine hydrochloride* (58).**


White solid, 59% yield. ^1^H NMR (400 MHz, DMSO) δ 9.90 (brs, 2H, N*H*_2_), 9.03 (s, 1H), 8.91 (d, *J* = 1.7 Hz, 1H), 8.81 (dd, *J* = 5.6, 1.7 Hz, 1H), 8.49 (dt, *J* = 8.0, 1.7 Hz, 1H), 7.98 (dd, *J* = 8.0, 5.6 Hz, 1H), 7.79 (d, *J* = 7.6 Hz, 2H), 7.56 (t, *J* = 7.6 Hz, 2H), 7.36 (t, *J* = 7.6 Hz, 1H), 7.27 (d, *J* = 1.7 Hz, 1H), 7.17 (dd, *J* = 8.1, 1.7 Hz, 1H), 7.04 (d, *J* = 8.1 Hz, 1H), 6.10 (s, 2H), 4.21 (t, *J* = 5.5 Hz, 2H), 3.37 (m, 2H), 3.30 (m, 2H). ^13^C NMR (101 MHz, DMSO) δ 152.0, 148.5, 148.4, 146.5, 143.6, 141.6, 140.1, 137.9, 131.3, 130.7, 127.7, 127.5, 126.8, 123.1, 119.3, 112.9, 109.5, 109.4, 102.2, 47.2, 42.0, 29.2. UPLC/MS (ESI) *m/z* = 399.3 [M+H]^+^.


**
*N-((3-(benzo[d]*
**
*[1,3]*
***dioxol-5-yl)-1-phenyl-1H-pyrazol-4-yl)methyl)-2-(pyridin-2-yl)ethan-1-amine hydrochloride* (59).**


White solid, 48% yield. ^1^H NMR (400 MHz, DMSO) δ 9.82 (brs, 2H, N*H*_2_), 8.96 (s, 1H), 8.73 (d, *J* = 5.5 Hz, 1H), 8.42–8.14 (m, 1H), 7.81 (m, 3H), 7.77–7.67 (m, 1H), 7.56 (t, *J* = 7.9 Hz, 2H), 7.37 (t, *J* = 7.9 Hz, 1H), 7.28 (s, 1H), 7.19 (d, *J* = 8.1 Hz, 1H), 7.04 (d, *J* = 8.1 Hz, 1H), 6.10 (s, 2H), 4.27 (t, *J* = 5.7 Hz, 2H), 3.47 (m, 4H); ^13^C NMR (101 MHz, DMSO) δ 154.8, 152.4, 148.9, 148.8, 145.5, 144.4, 140.3, 131.5, 131.1, 128.2, 127.3, 126.9, 125.6, 123.3, 119.7, 112.7, 109.9, 109.6, 102.6, 46.1, 42.2, 31.6. UPLC/MS (ESI) *m/z* = 399.3 [M+H]^+^.


**
*4-(4-(((3-(benzo[d]*
**
*[1,3]*
***dioxol-5-yl)-1-phenyl-1H-pyrazol-4-yl)methyl)amino)phenoxy)-N-methylpicolinamide* (60).**


Brown solid, 9% yield. ^1^H NMR (400 MHz, DMSO) δ 8.77 (q, *J* = 5.1 Hz, 1H, N*H*), 8.50 (s, 1H), 8.45 (d, *J* = 5.6 Hz, 1H), 7.83 (d, *J* = 8.1 Hz, 2H), 7.50 (t, *J* = 8.1 Hz, 2H), 7.37–7.24 (m, 4H), 7.08 (dd, *J* = 5.6, 2.6 Hz, 1H), 7.00 (d, *J* = 8.1 Hz, 1H), 6.96 (d, *J* = 8.6 Hz, 2H), 6.76 (d, *J* = 8.6 Hz, 2H), 6.03 (m, 3H, 1H N*H*), 4.21 (m, 2H), 2.77 (d, *J* = 5.1 Hz, 3H); ^13^C NMR (101 MHz, DMSO) δ 167.7, 164.8, 153.2, 151.3, 151.1, 148.5, 148.1, 147.8, 144.2, 140.4, 130.5, 130.0, 127.8, 127.0, 122.6, 122.2, 119.3, 119.0, 114.7, 114.4, 109.5, 109.3, 108.6, 102.1, 39.5, 26.9. UPLC/MS (ESI) *m/z* = 520.3 [M+H]^+^.


**
*N-((3-(benzo[d]*
**
*[1,3]*
***dioxol-5-yl)-1-phenyl-1H-pyrazol-4-yl)methyl)-2-morpholinoethan-1-amine dihydrochloride* (61).**


White solid, 45% yield. ^1^H NMR (400 MHz, DMSO) δ 11.31 (brs, 1H, N*H*), 9.87 (brs, 2H, N*H*_2_), 8.93 (s, 1H), 7.80 (d, *J* = 7.7 Hz, 2H), 7.57 (d, *J* = 7.7 Hz, 2H), 7.38 (t, *J* = 7.7 Hz, 1H), 7.30 (d, *J* = 1.7 Hz, 1H), 7.20 (dd, *J* = 8.0, 1.7 Hz, 1H), 7.05 (d, *J* = 8.0 Hz, 1H), 6.10 (s, 2H), 4.28 (m, 2H), 3.99 (m, 2H), 3.81 (m, 2H), 3.55 (m, 6H), 3.16 (m, 2H); ^13^C NMR (101 MHz, DMSO) δ 151.8, 148.6, 148.5, 140.1, 131.0, 130.8, 127.8, 126.7, 123.1, 119.3, 112.8, 109.5, 109.4, 102.3, 64.2, 52.9, 52.5, 42.3, 41.4. UPLC/MS (ESI) *m/z* = 407.3 [M+H]^+^.


***N-((3-(4-methoxyphenyl)-1-phenyl-1H-pyrazol-4-yl)methyl)-2-(pyridin-4-yl)ethan-1-amine hydrochloride* (62).**


Pale brown solid, 55% yield. ^1^H NMR (400 MHz, DMSO) δ 9.88 (brs, 2H, N*H*_2_), 9.00 (s, 1H), 8.84 (d, *J* = 5.9 Hz, 2H), 7.94 (d, *J* = 5.9 Hz, 2H), 7.80 (d, *J* = 7.7 Hz, 2H), 7.64 (d, *J* = 8.8 Hz, 2H), 7.56 (t, *J* = 7.7 Hz, 2H), 7.37 (t, *J* = 7.7 Hz, 1H), 7.06 (d, *J* = 8.8 Hz, 2H), 4.24 (m, 2H), 3.82 (s, 3H), 3.37 (m, 4H); ^13^C NMR (101 MHz, DMSO) δ 160.4, 158.4, 152.1, 142.5, 140.2, 131.3, 130.8, 130.5, 128.3, 127.6, 125.3, 119.3, 115.2, 112.8, 56.2, 46.4, 42.1, 32.2. UPLC/MS (ESI) *m/z* = 385.3 [M+H]^+^.


***N-((3-(4-methoxyphenyl)-1-phenyl-1H-pyrazol-4-yl)methyl)-2-(pyridin-3-yl)ethan-1-amine hydrochloride* (63).**


Pale brown solid, 43% yield. ^1^H NMR (400 MHz, DMSO) δ 9.95 (brs, 2H, N*H*_2_), 9.05 (s, 1H), 8.92 (d, *J* = 2.0 Hz, 1H), 8.81 (dd, *J* = 5.6, 2.0 Hz, 1H), 8.51 (d, *J* = 8.2 Hz, 1H), 8.00 (dd, *J* = 8.2, 5.6 Hz, 1H), 7.79 (d, *J* = 7.9 Hz, 2H), 7.64 (d, *J* = 8.7 Hz, 2H), 7.55 (t, *J* = 7.9 Hz, 2H), 7.36 (t, *J* = 7.9 Hz, 1H), 7.06 (d, *J* = 8.7 Hz, 2H), 4.22 (m, 2H), 3.82 (s, 3H), 3.45–3.26 (m, 4H); ^13^C NMR (101 MHz, DMSO) δ 160.4, 152.1, 146.8, 143.3, 141.3, 140.2, 138.1, 131.3, 130.7, 130.4, 127.6, 127.6, 125.3, 119.2, 115.1, 112.8, 56.2, 47.2, 42.1, 29.2. UPLC/MS (ESI) *m/z* = 385.3 [M+H]^+^.


***N-((3-(4-methoxyphenyl)-1-phenyl-1H-pyrazol-4-yl)methyl)-2-(pyridin-2-yl)ethan-1-amine hydrochloride* (64).**


Pale brown solid, 10% yield. ^1^H NMR (400 MHz, DMSO) δ 9.92 (brs, 2H, N*H*_2_), 8.99 (s, 1H), 8.75 (d, *J* = 7.9 Hz, 1H), 8.34 (t, *J* = 7.9 Hz, 1H), 7.87 (d, *J* = 7.9 Hz, 1H), 7.80 (m, 3H), 7.66 (d, *J* = 8.3 Hz, 2H), 7.56 (t, *J* = 7.8 Hz, 2H), 7.37 (t, *J* = 7.8 Hz, 1H), 7.06 (d, *J* = 8.3 Hz, 2H), 4.25 (t, *J* = 5.7 Hz, 2H), 3.82 (s, 3H), 3.50 (m, 4H); ^13^C NMR (101 MHz, DMSO) δ 152.7, 152.4, 144.5, 141.2, 132.7, 130.1, 122.0, 121.8, 121.0, 119.0, 118.2, 116.5, 114.5, 111.6, 111.5, 106.5, 47.2, 41.2, 35.7, 29.7. UPLC/MS (ESI) *m/z* = 385.3 [M+H]^+^.


***3-(4-methoxyphenyl)-1-phenyl-4-(pyrrolidin-1-ylmethyl)-1H-pyrazole oxalate* (65).**


White solid, 64% yield. ^1^H NMR (400 MHz, DMSO) δ 8.75 (s, 1H), 7.87 (d, *J* = 7.3 Hz, 2H), 7.66 (d, *J* = 8.0 Hz, 2H), 7.54 (t, *J* = 7.3 Hz, 2H), 7.35 (t, *J* = 7.3 Hz, 1H), 7.06 (d, *J* = 8.0 Hz, 2H), 4.34 (s, 2H), 3.82 (s, 3H), 3.16 (m, 4H), 1.87 (m, 4H); ^13^C NMR (101 MHz, DMSO) δ 165.4, 160.3, 152.4, 140.1, 131.1, 130.6, 130.4, 127.5, 125.5, 119.3, 115.1, 112.9, 56.1, 53.6, 48.6, 23.6. UPLC/MS (ESI) *m/z* = 334.2 [M+H]^+^.


***4-((3-(4-methoxyphenyl)-1-phenyl-1H-pyrazol-4-yl)methyl)morpholine hydrochloride* (66).**


Yellow solid, 62% yield. ^1^H NMR (400 MHz, DMSO) δ 11.11 (brs, 1H, N*H*), 8.93 (s, 1H), 7.84 (d, *J* = 7.9 Hz, 2H), 7.66 (d, *J* = 8.3 Hz, 2H), 7.56 (t, *J* = 7.9 Hz, 2H), 7.39 (d, *J* = 7.9 Hz, 1H), 7.07 (d, *J* = 8.3 Hz, 2H), 4.53–4.29 (m, 2H), 4.01–3.69 (m, 7H), 3.34 (m, 2H), 3.11–2.91 (m, 2H); ^13^C NMR (101 MHz, DMSO) δ 160.4, 153.1, 140.1, 132.5, 130.7, 127.8, 125.2, 119.4, 115.1, 109.9, 64.0, 56.1, 61.4, 50.8. UPLC/MS (ESI) *m/z* = 350.3 [M+H]^+^.


***N-((3-(3,4-dimethoxyphenyl)-1-phenyl-1H-pyrazol-4-yl)methyl)-2-(4-fluorophenyl)ethan-1-amine hydrochloride* (67).**


White solid, 70% yield. ^1^H NMR (400 MHz, DMSO) δ 9.55 (brs, 2H, N*H*_2_), 8.89 (s, 1H), 7.83 (m, 2H), 7.61–7.51 (m, 2H), 7.41–7.34 (m, 1H), 7.32–7.26 (m, 2H), 7.25 (d, *J* = 2.0 Hz, 1H), 7.21 (dd, *J* = 8.2, 2.0 Hz, 1H), 7.17–7.13 (m, 2H), 7.07 (d, *J* = 8.2 Hz, 1H), 4.24 (s, 2H), 3.83 (s, 3H), 3.82 (s, 3H), 3.20 (s, 2H), 3.07–2.93 (m, 2H); ^13^C NMR (101 MHz, DMSO) δ 162.1 (d, ^1^*J*_C-F_ = 241 Hz), 152.4, 150.1, 149.8, 140.2, 134.3 (d, ^4^*J*_C-F_ = 3 Hz), 131.5 (d, ^3^*J*_C-F_ = 8 Hz), 131.2, 130.7, 127.7, 125.5, 121.6, 119.3, 116.3 (d, ^2^*J*_C-F_ = 21 Hz), 113.0, 112.8, 112.7, 56.6, 56.5, 48.4, 41.9, 31.5. UPLC/MS (ESI) *m/z* = 432.3 [M+H]^+^.


***4-(((4-fluorobenzyl)oxy)methyl)-3-(4-methoxyphenyl)-1-phenyl-1H-pyrazole* (68).**


Pale yellow, 47% yield. ^1^H NMR (400 MHz, DMSO) δ 8.62 (s, 1H), 7.89 (d, *J* = 8.0 Hz, 2H), 7.76 (d, *J* = 8.3 Hz, 2H), 7.51 (t, *J* = 8.0 Hz, 2H), 7.43 (m, 2H), 7.31 (t, *J* = 8.0 Hz, 1H), 7.20 (m, 2H), 7.02 (d, *J* = 8.3 Hz, 2H), 4.57 (s, 2H), 4.52 (s, 2H), 3.80 (s, 3H); ^13^C NMR (101 MHz, DMSO) δ 162.6 (d, ^1^*J*_C-F_ = 242 Hz), 160.1, 151.9, 140.4, 135.3 (d, ^4^*J*_C-F_ = 3 Hz), 130.9 (d, ^3^*J*_C-F_ = 8 Hz), 130.7, 130.5, 129.6, 127.1, 126.1, 119.1, 118.0, 116.0 (d, ^2^*J*_C-F_ = 21 Hz), 115.0, 71.3, 63.1, 56.1. UPLC/MS (ESI) *m/z* = 389.2 [M+H]^+^.


***N-((3-(2,3-dihydrobenzo[b][1,4]dioxin-6-yl)-1-(pyridin-2-yl)-1H-pyrazol-4-yl)methyl)-2-(pyrrolidin-1-yl)ethan-1-amine dihydrochloride* (69).**


Brown solid, 10% yield. ^1^H NMR (400 MHz, DMSO+D_2_O) δ 8.97 (s, 1H), 8.49 (d, *J* = 6.1 Hz, 1H), 8.06–7.92 (m, 2H), 7.39 (t, *J* = 6.1 Hz, 1H), 7.24–7.11 (m, 2H), 6.99 (d, *J* = 8.1 Hz, 1H), 4.40–4.18 (m, 6H), 3.62 (m, 2H), 3.55–3.37 (m, 4H), 3.03 (m, 2H), 2.10–1.77 (m, 4H). ^13^C NMR (101 MHz, DMSO+D_2_O) δ 152.9, 151.5, 149.8, 145.4, 144.9, 141.3, 130.3, 126.1, 124.0, 122.7, 118.9, 118.0, 113.5, 113.3, 65.6, 65.5, 55.0, 50.8, 43.9, 43.0, 23.9. UPLC/MS (ESI) *m/z* = 406.4 [M+H]^+^.

The ^1^H and ^13^C NMR spectra of compounds **13**–**69**; UPLC chromatograms and HRMS spectra of **13**, **14**, **17**, **22**, **26**, **29**, **33**, **35**, **38**, **41**, **44**, **48**, **49**, **54**, **68** and **69** are available in Appendix A.

### 3.2. Biology

#### 3.2.1. Parasite and Cell Cultures

The MRC-5_SV2_ human lung fibroblast cell line was cultured in MEM + Earle’s salts-medium, supplemented with 2 mM L-glutamine, 16.5 mM NaHCO_3_, and 5% inactivated fetal calf serum. For evaluating the antileishmanial activity, the *L. infantum* MHOM/MA(BE)/67 strain was used. This strain was maintained in a Golden hamster (*Mesocricetus auratus*). Amastigotes were collected from the spleen of an infected donor hamster using differential centrifugation. Primary peritoneal mouse macrophages were used as host cells and were collected 2 days after peritoneal stimulation with a 2% potato starch suspension. For the intracellular *T. cruzi* assays, the Tulahuen CL2 strain was used that expresses β-galactosidase. The strain was maintained in MRC-5_SV2_ cells in MEM medium. The *T. b. brucei* Squib 427 strain or *T. b. rhodesiense* STIB-900 strain was used for screening against the extracellular African trypanosomes. The strains were maintained in Hirumi’s Modified Iscove’s medium-9 (HMI-9), supplemented with 10% inactivated fetal calf serum. All cultures and assays were performed at 37 °C with 5% CO_2_.

#### 3.2.2. Compound Solutions/Dilutions

Compound stock solutions were prepared in 100% DMSO at 20 mM. The compounds were serially pre-diluted by robotic plate production (4-fold) in DMSO followed by a further (intermediate) dilution in demineralized water to assure a final in-test DMSO concentration of <1%. Identical test plates containing 10 µL of the different concentrations of each compound were produced for the different assays. For the cytotoxicity assays, 5 concentrations per compound were tested (64–16–4–1–0.25 µM). For the parasitic assays, 10 concentrations per compound were tested, ranging from 64 µM to 0.2 nM.

#### 3.2.3. Drug Sensitivity Assays

##### Cytotoxicity Assays

Assays were performed in sterile 96-well microtiter plates, each well containing 10 µL of the watery compound dilutions together with 190 µL of MRC-5_SV2_ inoculum (1.5 × 10^5^ cells/mL). Cell growth was compared to untreated control wells (100% cell growth) and medium-control wells (0% cell growth). After 3 days incubation, MRC-5 viability was assessed fluorimetrically after addition of 50 μL resazurin per well (final concentration of 10 µg mL^−1^). After 4 h at 37 °C, fluorescence was measured (λ_ex_ 550 nm, λ_em_ 590 nm). The results were expressed as % reduction in cell growth/viability compared to positive (100% growth–0% inhibition) and negative (0% growth–100% inhibition) control wells from which EC_50_ values (50% effective concentrations) were derived using an excel macro that calculates the interpolation of the concentrations corresponding to the flanking points, just above and just below 50% inhibition.

##### *T. cruzi* Assays

MRC-5 cells were infected in 96-well plates using a harvested *T. cruzi* inoculum (4 × 10^3^ MRC-5 cells/well + 4 × 10^4^ parasites/well) and exposed to 10 µL of the compound dilutions (200 µL/well total volume). Parasite growth determined after 7 days of incubation and compared to untreated infected controls following the addition of the substrate CPRG (chlorophenolred ß-D-galactopyranoside): 50 µL/well of a stock solution containing 15.2 mg CPRG + 250 µL Nonidet in 100 mL PBS. The absorbance was measured spectrophotometrically at 540 nm after 4 h incubation at 37 °C and EC_50_ values were calculated.

##### *L. infantum* Assays

Primary peritoneal mouse macrophages (PMMs) were prepared in RPMI-1640 medium, supplemented with 2 mM L-glutamine, 16.5 mM NaHCO_3_, and 5% inactivated fetal calf serum. The macrophages are infected after 48 h with spleen-derived *L. infantum* amastigotes (3 × 10^4^ PMM + 4.5 × 10^5^ parasites/well). The compounds were added after 2 h of infection. After 5 days incubation, parasite burdens (mean number of amastigotes/macrophage) were assessed microscopically after staining with a 10% Giemsa solution and EC_50_ values were calculated.

##### *T. brucei* and *T. b. rhodesiense* Assays

Parasite suspensions (1.5 × 10^4^ *T. b. brucei*/well or 4 × 10^3^ *T. b. rhodesiense*/well) in HMI-9 medium were exposed in a total volume of 200 µL to 10 µL of the compound dilutions Parasite growth was assessed fluorimetrically after 3 days of incubation by the addition of 50 µL resazurin per well. After 6 h (*T. b. rhodesiense*) or 24 h (*T. b. brucei*) at 37 °C, fluorescence was measured (λ_ex_ 550 nm, λ_em_ 590 nm), from which viability and EC_50_ values could be determined. 

## 4. Conclusions

A series of 1,3-diarylpyrazoles based on the ‘hit’ 26/HIT 8 was synthesized and cytotoxicity evaluated against the MRC-5 fibroblast cell line and primary peritoneal mouse macrophages. Several compounds showed moderate to high cytotoxicity. Most notable were molecules containing a basic, *N*-substituted aminomethyl-R group. For the non-cytotoxic compounds, the potency was investigated against a range of human pathogenic protozoa, including *T. cruzi*, *T. b. rhodesiense*, and *L. infantum*. Compound **22** showed the most potent antiparasitic activity of the series, specifically against *T. b. rhodesiense* (EC_50_ of 0.74 μM, selectivity index of 86). Some other compounds (**14**, **21**, **26**/HIT **8**, and **41**) displayed modest but broad activity against all pathogens tested. We are currently investigating the autophagy inducing properties of the non-toxic series reported here.

## Data Availability

Data are contained within the article.

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
