# Peer review of "Design and Synthesis of 1,3-Diarylpyrazoles and Investigation of Their Cytotoxicity and Antiparasitic Profile"

_ijms, 2024, doi:10.3390/ijms25094693_

Round 1

Reviewer 1 Report

Comments and Suggestions for Authors

The article by Murat Bozdag et al., entitled “Design and synthesis of 1,3-diarylpyrazoles and investigation 2 of their cytotoxicity and antiparasitic profile,” is devoted to the synthesis and study of the antiparasitic activities of new 1,3-diarylpyrazoles. The paper in its current form cannot be published in IJMS for one simple reason, namely, not a single compound has been fully characterized. Despite the fact that the compounds are solid, no melting points of new compounds are indicated, nor is there a comparison of the melting points of the compounds obtained in the work with previously synthesized compounds. Signals in 1H-NMR and 13C-NMR spectra are not assigned. If the UPLC/MS method was used to prove the purity of the obtained compounds, then in the corresponding part (Supplementary Material, which is missing), it is necessary to provide chromatograms of the studied samples. In this case, primary instrument data are necessary because the reviewer is not confident that the authors obtained the target compounds in pure form. To prove this, it is necessary to provide elemental analysis data, or at least HRMS. Accordingly, if there is no confidence in the purity of the resulting compounds, this calls into question the results of biological tests. If the authors decide to rework the article, then I would like to give them advice, not to try to synthesize as many compounds as possible, but to work out methods in order to increase the yields of target compounds, and perhaps divide the material into several articles. It is strange to see yields of 7% in the reductive amination reaction of 4-formyl-1,3-diarylpyrazoles, while much higher yields were obtained in this reaction with more "complex" objects. (10.1016/j.jorganchem.2009.08.013, 10.1142/S1088424612501246)

Author Response

1H and 13C Spectra of compounds 13-69 and UPLC and HRMS reports of compounds 13, 14, 17, 22, 26, 29, 33, 35, 41, 44, 48, 49,54, 68 and 69 are reported in the supporting information section. We are confident that the experimental data reported in our manuscript align with the journal’s requirements. The reviewer has expressed concerns regarding the low yield reported for compound 14. Some of the derivatives we reported in our work indeed afforded in low yields due to lose of significant quantities during purification steps, however it was necessary to have pure products. While we were focused on obtaining the pure products, the low milligram quantities were acceptable for us. Therefore, we did not repeat the reactions to increase yields and quantities.

Reviewer 2 Report

Comments and Suggestions for Authors

The manuscript by Veken et al described the synthesis of library compounds and studied their various activities.  The manuscript is well-written except for some minor typos and mistakes. My major concern is the lack of supporting information to assess the purity of the compounds.

Comments on the Quality of English Language

The manuscript by Veken et al described the synthesis of library compounds and studied their various activities.  The manuscript is well-written except for some minor typos and mistakes. My major concern is the lack of supporting information to assess the purity of the compounds.

Author Response

The Supporting Information file is provided with 1H and 13C spectra and UPLC and HRMS results of compounds with HIT8/26 and 29 with compounds 13, 14, 17, 22, 33, 35, 41, 44, 48, 49,54, 68 and 69.

Reviewer 3 Report

Comments and Suggestions for Authors

ijms-2911141, Design and synthesis of 1,3-diarylpyrazoles and investigation of their cytotoxicity and antiparasitic profile

Overall, the study is very strong and correctly performed, but several aspects warrant further discussion and clarification to enhance its scientific rigor and impact.

The introduction is very generally describing the pyrazole ring, giving example of its use in various types of compounds. Most of them are not related to the action in hand. I consider pertinent to the discussion only the mention of the protein kinases (the authors should correct kinases to protein kinases on row 37). The inhibition of PKs is known to be a target for protozoa. See for example:

1.     Systematic functional analysis of Leishmania protein kinases identifies regulators of differentiation or survival

2.     In vitro and in vivo evaluation of kinase and protease inhibitors against Trypanosoma evansi

The aforementioned article even cites a pyrazole inhibitor as antiparasitic drug.

I would change the introduction on this basis of PKI and their impact on Trypanosoma and Leishmania. I would delete all the mention of lonazolac (in my view the acetic acid fragment is the true active part) and the whole section on rows 38-44. The figure 1 should also be changed to better reflect pyrazole compounds that inhibit PKs and have antiparasitic effects.

On row 106, the authors use the compounds illustrated in Figure 1 as a proof of the importance of the diarylpyrazole scaffold. Looking on this matter I found a better proof, an article (Quantitative and Qualitative Analysis of the Anti-Proliferative Potential of the Pyrazole Scaffold in the Design of Anticancer Agents) that demonstrated that 1,3-diphenyl-pyrazole is a useful scaffold for potent and targeted anticancer candidates. I think the authors should refer on this paper and correlate the anticancer properties with the PK inhibition and with the potential for antiparasitic effects.

The title mention “Design”, but the article does not reflect it. I’m sure the authors had a strategy and I consider that it should be shared with the readers in a dedicated section of the paper.

The manuscript has a strong synthesis section, but the medicinal section feels underdeveloped. Considering the journal, I feel that the authors should develop better the structure-activity discussion and point out the future direction of lead optimization. The article could be improved with more graphical representations.

I think that the methods are described very summarily in terms of the biological assays and that should be improved. Please add details as how many points were used for EC50 calculation, the interpolation method and if available a measure of error for the data.

I think the authors should consider not only the toll-like receptor hypothesis, but also the possibility of the compounds to inhibit one or several relevant PK.

Comments on the Quality of English Language

the language is very good, but there are editing mistakes

Round 2

Reviewer 1 Report

Comments and Suggestions for Authors

The article by Murat Bozdag et al., entitled “Design and synthesis of 1,3-diarylpyrazoles and investigation 2 of their cytotoxicity and antiparasitic profile,” is devoted to the synthesis and study of the antiparasitic activities of new 1,3-diarylpyrazoles. The article, even in its current revised form, cannot be published in IJMS for one simple reason, namely, not a single compound has been fully characterized (see previous report). For all solid compounds there are no melting points, one of the main characteristics of the resulting substance. In addition, there is no elemental analysis data. Spectra and chromatograms in Supplementary Material cannot be evidence of the purity of the compounds obtained, since in NMR spectra impurities of 5% or less are usually not visible, and in Chromatograms, UV detection allows us to judge only the presence of “organic” compounds, and the mass detector only indicates the presence of substances capable of ionization, and only under these conditions. Moreover, usually the chromatogram (whether mass or UV detection) of a pure compound presents a peak close to a Gaussian distribution, which we do not observe in the chromatograms of compounds 17,22, 26, 29, 35, 38, 41, 44, 48. 49, 54, 68, 69. In addition, a previous report pointed out the need to assign all signals in both 1H-NMR and 13C-NMR spectra. This was not done, which leads to the fact that, for example, in compound 27 there are 21 different carbon atoms, and only 20 are described in the 13C-NMR spectrum, although there are 21 signals in the spectrum itself. Based on the above, the article should not be accepted for publication, and the attitude of the authors towards the comments and wishes of the reviewers does not allow the revised version to be considered for publication in the future.

Reviewer 2 Report

Comments and Suggestions for Authors

The authors have addressed my concerns and the manuscript is ready to be accepted for publication.

Comments on the Quality of English Language

The authors have addressed my concerns and the manuscript is ready to be accepted for publication.

Author Response

We thank to the reviewer 2.

Round 3

Reviewer 1 Report

Comments and Suggestions for Authors

Dear scientific editor and authors of the article!

There is a substitution of concepts here. In my review, I wrote about proof of the purity of the obtained substances (the presence of melting points for solid compounds and elemental analysis data). Instead, authors appeal to the journal's rules. NMR spectra cannot indicate the purity of the compounds, since they do not show impurities less than 5%. The absence of absorption bands in the UV spectrum does not prove anything, since the impurity may not absorb in this region, which is what we observe for compounds 17, 22, 26, 29, 35, 38, 41, 44, 48, 49, 54, 68 , 69, that is, almost all for which mass spectra are given. In addition, it is clear that even in the UV spectra, where the purity was determined to be 95 and 100%, other peaks are present (compounds 13, 17, 22, 26, 29, 48, 49, 54, 68, 69) Absence of signals in mass spectra may be due to the fact that the impurity is not ionized under the experimental conditions. Thus, the purity of the compounds has not been proven; therefore, the further part of the work related to the determination of bioactivity is to some extent meaningless, because it is not clear what activity the authors are studying, target compounds or impurities. Based on this, it is not possible to draw any conclusions about the structure-activity relationship.

As for the authors' remark that Reviewer takes decisions based on a perceived attitude of authors, and not on scientific arguments, this is not true. The article simply does not provide evidence of the purity of the compounds obtained.

Author Response

We have responded to the comments of the Academic Editor.